# TokenButler: Token Importance is Predictable

## Abstract

Large Language Models (LLMs) rely on the Key-Value (KV) Cache to store token history, enabling efficient decoding of tokens. As the KV-Cache grows, it becomes a major memory and computation bottleneck, however, there is an opportunity to alleviate this bottleneck, especially because prior research has shown that only a small subset of tokens contribute meaningfully to each decoding step. A key challenge in finding these *critical tokens* is that they are dynamic, and heavily input query-dependent. Existing methods either risk quality by evicting tokens permanently, or retain the full KV-Cache but rely on retrieving chunks (pages) of tokens at generation, failing at dense, context-rich tasks. Additionally, many existing KV-Cache sparsity methods rely on inaccurate proxies for token importance. To address these limitations, we introduce **TokenButler**, a high-granularity, query-aware predictor that learns to identify these critical tokens. By training a light-weight predictor with less than $1.2\%$ parameter overhead, TokenButler prioritizes tokens based on their contextual, predicted importance. This improves perplexity & downstream accuracy by upto $8\%$ relative to SoTA methods for estimating token importance. We evaluate TokenButler on a novel synthetic small-context co-referential retrieval task, demonstrating near-oracle accuracy. Furthermore, we show that TokenButler minimizes the gap to the oracle throughput and outperforms prior methods by up to $3\times$. Code, models, dataset and benchmarks `are available`.

## 1 Introduction

As Large Language Models (LLMs) become more widely used (Thoppilan et al., 2022; Yuan et al., 2022; Wei et al., 2022; Zhang et al., 2023a), recent advances have extended their context lengths to 128k–1M tokens. However, recent research on long-context evaluation (Vodrahalli et al., 2024) reveal that model quality degrades noticeably as early as 8k tokens, even without token compression. Furthermore, as input sequences grow, the memory footprint of the Key-Value (KV) cache, which stores intermediate key-value pairs to skip recomputation, scales linearly. This increases memory requirements and stresses the memory-bandwidth, and raises important questions on how effectively existing token-pruning techniques address KV-cache size, especially in context-dense downstream tasks that go beyond retrieval or summarization. There have been several efforts at improving model quality while addressing KV-cache memory issues. Certain transformer variants aim at implicitly compressing the KV-cache via sparsity, quantization, efficient-attention, or low-rank compression (Child et al., 2019; Choromanski et al., 2020; Katharopoulos et al., 2020; Shazeer, 2019; Pope et al., 2022; Sun et al., 2024; Akhauri et al., 2024b; Chen et al., 2025).

The current literature on token pruning addresses this growing memory footprint in three ways. **(1)** *Purely static strategies* limiting KV-Cache to a fixed budget with fixed rules on removing tokens, naturally reducing bandwidth and storage (StreamingLLM (Xiao et al.), and Sliding Window Attention (Luong, 2015)), **(2)** *Adaptive strategies* that permanently sacrifice *less important* past-tokens effectively fixing the memory and bandwidth footprint ($H_2O$, SnapKV (Zhang et al., 2023b; Li et al., 2024)), and **(3)** *Adaptive dynamic strategies* that preserve the entire KV-Cache but access only a subset of the Key-Value entries (the *more important* past-tokens), incurring higher memory (storage) cost, but reducing memory bandwidth (accesses to memory) during the decode stage (generation) (Quest, FastGen, (Tang et al., 2024; Ge et al.))

Each of these strategies to limit storage and bandwidth costs have implications. Specifically, token preference has been shown to be highly dependent on the query (Tang et al., 2024), and vary significantly at generation. Purely static strategies do not have any query-awareness, and will fail

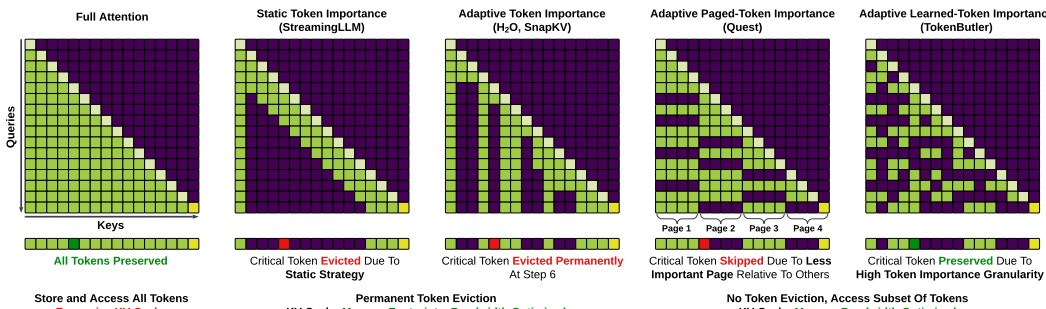

Figure 1: Full-Attention preserves all tokens, enabling access to the critical token (dark green) during the last decode step. Static strategies like StreamingLLM will not be able to access this token. Methods like $H_2O$ may have evicted the token at an earlier decode step, if deemed unimportant. Paged-Token importance may cause a page-miss of a critical token in context dense tasks. **TokenButler** can effectively predict critical tokens, and can be leveraged by existing methods to offer both high-granularity and cheap importance estimation.

at retrieving contextually relevant tokens. Additionally, *Adaptive strategies* that have permanently discarded tokens deemed less important at a prior decode step will not be able to fetch relevant tokens if the course of discussion is *co-referential* (Vodrahalli et al., 2024). A conversation is co-referential if text introduced earlier is referenced again later, requiring accurate retrieval and reasoning over the earlier reference. For co-referential conversations *adaptive dynamic strategies* is the most reasonable solution. Current methods rely on *token grouping* to make the dynamic calculation of token relevance efficient (Tang et al., 2024).

There are several *metrics* to quantify token importance including recency, aggregate attention scores, and others listed in Table 1. Token Sparsity methods use these metrics to guide token eviction or retrieval decisions. There is an important interplay between methods and metrics. Some methods permanently evict tokens based on strong metrics like the attention score. However, evicted tokens may become relevant later during generation. Other methods preserve tokens but selectively retrieve a subset during generation. These methods cannot rely on strong metrics such as attention scores. This is because only a subset of the KV-cache is fetched during generation based on a token importance metric and that metric cannot be the result of the computation itself (attention score). To address this, we propose a novel *learned* metric of token importance, called **TokenButler**, which provides fine-granularity estimates of token importance. Our contributions are summarized as:

- We train a light-weight predictor ($< 1.2\%$ parameter overhead) for estimating token-importance, achieving up to 75% accuracy in identifying the top 50% tokens.

- We introduce a synthetic, co-referential decode benchmark that demonstrates where current KV-cache sparsity techniques break by either evicting or overlooking context-critical tokens. On this benchmark, TokenButler preserves critical tokens with near-oracle accuracy while still achieving aggressive KV-cache sparsity.

- TokenButler improves the wikitext perplexity and downstream accuracy over existing token sparsity metrics by over $8\%$, identifying critical tokens with *near-oracle* accuracy.

- We show that TokenButler achieves up to $3\times$ better throughput than recent token importance estimation methods like TokenSelect (Wu et al., 2024).

## 2 RELATED WORK

Prior work has shown that transformers exhibit very strong contextual behavior, where head and neuron importance heavily depends on the query. (Liu et al., 2023; Akhauri et al., 2024a) leverage this behavior to contextually prune entire neurons and heads on a per-query basis. These methods train small neural networks to predict the relative importance–quantified using parameter magnitudes or gradients of neurons across the transformer. This *magnitude* can be considered as the *metric* of contextual importance. Furthermore, these works explore techniques of using these metrics to then

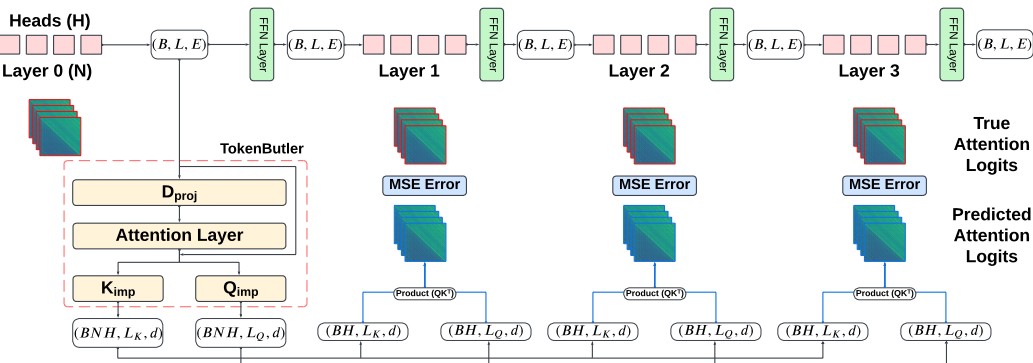

Figure 2: TokenButler is a light-weight predictor, with a down projection $D_{\mathrm{proj}}$ for cheaper attention, attention layer, and Key-Query projection *neural networks*. These $\{Q_{\mathrm{imp}}, K_{\mathrm{imp}}\}$ effectively map the output of the attention mechanism to $N \times H$ Key-Query projection tensors (N: Num. Layers, H: Num. Heads) on a small interaction-dimension $d \ll E$. The full-(pre-softmax) attention logits can then be computed for every head across all layers by taking Product($QK^T$). At train-time, we minimize the MSE Error between true and prediction attention logits to learn the LLM behavior.

prune heads globally, or on a per-layer basis. This idea of pruning on a global or per-layer basis can be considered as the *method*, which leverages the *metric* to make decisions.

This contextual behavior applies to token importance by design, as the attention mechanism explicitly captures tokens relevant to a query. However, while *methods* to prune heads are simpler, as there is a fixed number of heads, *methods* to prune tokens are more expensive to realize. Specifically, for a transformer with $N$ layers and $H$ heads per-layer and $L$ past-tokens, every head has to decide which *subset $S$* of $L$ tokens are the most important at every decode step. This implies that any given *metric* has to be calculated for $N \times H \times L$ tokens, at *every decode step*.

| Method | Metric |
|---|---|
| StreamLLM | Recency-based sliding window |
| $H_2O$ | Attention Score for Token Eviction |
| SnapKV | Pooled Attention Score over a Fixed Window for Token Eviction |
| Quest | Query product with Per-Page Min–Max Token Magnitudes for Page Loading |
| TokenButler | Predicted Importance for Fine-Grained Token Loading |

Table 1: Metrics for token importance

As presented in Table 1, there have been significant efforts towards *co-designing* metrics with methods of token sparsity. The simplest methods are *purely static strategies*, StreamingLLM (Xiao et al.) relies on *recency* as a metric of token importance, with a sliding-window plus initial anchor tokens attention to fix a KV-Cache budget. More recently, methods like $H_2O$ (Zhang et al., 2023b) and SnapKV (Li et al., 2024) avoid naïve sparsification of tokens, and instead rely on attention scores to permanently evict low-importance tokens. This can be a major limitation when tasks require synthesizing or reasoning over information distributed across the context (Vodrahalli et al., 2024), as a token that becomes important later in the decoding stage may be evicted due to its low importance at the current step and low KV-Budget. To alleviate this issue, *Adaptive Dynamic Strategies* such as Quest (Tang et al., 2024) preserve all tokens, and dynamically decide which subset of tokens to fetch for a given query. Instead of calculating full attention scores to ensure the most important tokens are fetched (which can be prohibitively expensive), Quest relies on paging, preserving all tokens in paged memory, and selectively fetches important pages. To determine page importance, the dot product of query with min-max token values within a page is used as a proxy. This reduces memory bandwidth but does not optimize memory footprint. Furthermore, its sparsity is limited to the granularity of pages limiting its effectiveness in more challenging co-referential tasks as we will show. TokenSelect (Wu et al., 2024) also preserves all tokens and selects the important ones based on the dot product between queries and keys but it intelligently avoids doing that with every query based on the cosine similarity between different queries. However, this method incurs a high overhead due to the need of performing dot products with a high dimension.

While *metrics* that rely on attention scores are an effective way to estimate token importance, its usefulness is limited as it is tied to the *method*, necessitating token-eviction, or paged-token fetching, or a high overhead. By contrast, we propose to learn a lightweight token-importance predictor,

*TokenButler*, which cheaply approximates token-level attention logits using QK projections from the first layer of an LLM. This preserves fine-grained control over tokens (like full attention) while staying efficient: approximately 1% the size of the main LLM.

## 3 METHODOLOGY

We use a predictor to identify the most important tokens at each decode step. The predictor is designed to (1) use only the output of the first LLM layer to predict sparsity across all LLM layers thereby running efficiently and ahead-of-time, (2) be trained directly on minimizing error between its predicted attention maps and the LLM's actual attention maps. In this section, we describe the **TokenButler** predictor architecture and training methodology.

### 3.1 PREDICTOR DESIGN

TokenButler is a lightweight transformer ($\approx 1\%$ of the LLM size), depicted in Figure 2. For each layer and head, TokenButler estimates token-importance. The predictor takes in the hidden-states from the attention mechanism of the first layer, down-projects it, adds an attention layer to process the sequence, and passes it to a query and key (QK) projection neural network (QK-NN). These QK-NNs capture the behavior of all heads from later layers in the LLM.

Given hidden states $\mathbf{I} \in \mathbb{R}^{B \times L \times E}$ (batch $B$, length $L$, embedding $E$), the predictor applies an attention sub-network: a dimensionality-reduction projection (`Linear`) for efficient self-attention; one self-attention block over the reduced states to capture token context; and a feed-forward block that up-projects back to $E$ to produce $\mathbf{I}' \in \mathbb{R}^{B \times L \times E}$, which is added to $\mathbf{I}$ (residual). Next, TokenButler uses two projection networks $\{Q_{\text{imp}}, K_{\text{imp}}\}$ (each two linear layers with `SiLU`) to produce per-layer/per-head importance queries and keys from $\mathbf{I}'$, i.e., $\mathbf{Q}_{\text{imp}} = Q_{\text{imp}}(\mathbf{I}')$ and $\mathbf{K}_{\text{imp}} = K_{\text{imp}}(\mathbf{I}')$. Their outputs are reshaped to $\mathbb{R}^{B \times N \times H \times L \times d}$ (LLM layers $N$, heads $H$, head dimension $D$, interaction predictor-head dimension $d \ll E$), then $N$ and $H$ are flattened to yield $\mathbf{Q}_{\text{imp}}, \mathbf{K}_{\text{imp}} \in \mathbb{R}^{(BNH) \times L \times d}$. Approximate attention logits for each (layer, head, token) triplet use the scaled dot-product $\mathbf{A}_{\text{pred}} = \mathbf{Q}_{\text{imp}} \mathbf{K}_{\text{imp}}^{\mathsf{T}} / \sqrt{d} \in \mathbb{R}^{(BNH) \times L \times L}$; these unnormalized logits mimic the LLM's pre-softmax attention maps and, per layer and head, predict how strongly each token attends to every other token according to TokenButler's learned notion of token importance. We attach TokenButler at layer 0 so that all subsequent layers can be sparsified; attaching it deeper yields slightly higher recall but forces earlier layers to remain dense (Appendix §D).

### 3.2 PREDICTOR TRAINING

The LLM is frozen and we train only the TokenButler predictor. We run a forward pass of the LLM on the C4-*realnewslike* training corpus and extract its (pre-softmax) attention logits $\mathbf{A}_{\text{true}} \in \mathbb{R}^{(BNH) \times L \times L}$ *before* causal masking and softmax. Meanwhile, TokenButler produces its approximate logits $\mathbf{A}_{\text{pred}}$. We then minimize a mean-squared-error (MSE) loss between the two as $\mathcal{L}_{\text{MSE}} = ||\mathbf{A}_{\text{pred}} - \mathbf{A}_{\text{true}}||_2^2$. In practice, for each training batch:

1. **Forward pass** Compute $\mathbf{A}_{\text{true}}$ for each layer $n = 1, \ldots, N$ and head $h = 1, \ldots, H$, pass the first-layer output of the LLM to the predictor to obtain $\mathbf{A}_{\text{pred}}$.

2. **Loss computation.** Accumulate MSE across all layers (except the first layer) and heads.

3. **Backward update (predictor only).** Update TokenButler's parameters; the LLM remains frozen.

The predictor learns to approximate attention patterns of the full model with minimal overhead. In downstream usage, it can thus rapidly identify which tokens are most critical at per-token granularity, without performing expensive attention computations. Training overhead is modest: on a single A6000 GPU with a frozen base model, predictor training scales lightly from **7h17m (12.4M params)** to **8h42m (287M)**, since the base forward pass dominates (Appendix §E).

## 4 ACCURACY EVALUATION

We train and evaluate the accuracy of our predictors on Llama-3.2-3B, Llama-3.1-8B (Grattafiori et al., 2024), Llama-2-7b-hf (Touvron et al., 2023), Mistral-7B-v0.1 (Jiang et al., 2023), Phi-3.5-mini-instruct, and Phi-3-mini-4k-instruct (Abdin et al., 2024). The predictors are trained on the same

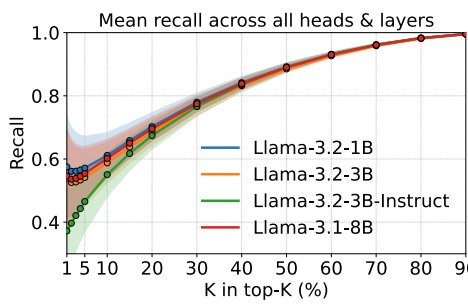

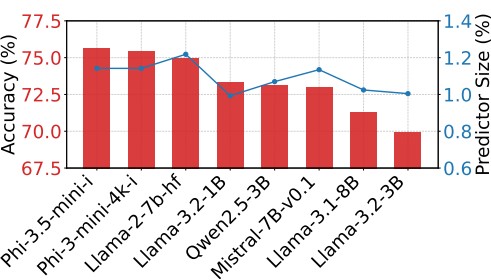

Figure 3: Top-K recall of TokenButler across Llama-1B/3B/8B and Llama-3B-Instruct variant. On average, TokenButler achieves 51% Recall@1%.

Figure 4: For predictors within a $[1, 1.2]\%$ parameter-count budget relative to the target LLM, the accuracy in identifying the top 50% most important tokens is between 70–75%.

text-corpus, resulting in 80-100M tokens (due to tokenizer differences) using C4-*realnewslike* (Raffel et al., 2019).

## 4.1 PREDICTOR ACCURACY & TOP-K RECALL

We evaluate TokenButler with two complementary metrics. *Token Classification Accuracy* treats importance prediction as a binary classification problem: given the LLM's attention map as ground truth, how often does the predictor correctly flag tokens in the top 50% importance set? Across models, TokenButler achieves 70–75% accuracy with only 1–1.2% parameter overhead (Figure 4).

*Top-K Recall* measures how well the predictor surfaces the most critical tokens under tight budgets: keeping only the predictor's top $K\%$ tokens, what fraction of the LLM-identified high-importance tokens are recovered? On WikiText2, averaged over Llama-1B/3B/8B and a Llama-3.2-3B-Instruct variant, TokenButler reaches $\sim$51% Recall@1% and improves steadily with larger $K$ (Fig. 3). This high top-K indicates the predictor reliably preserves the most informative token, which is required for more aggressive sparsity.

## 4.2 EVALUATION ON A SYNTHETIC TASK FOR TOKEN RETRIEVAL

We evaluate TokenButler on a difficult synthetic task inspired by Multi-Round Co-reference Resolution (Vodrahalli et al., 2024), using concise sequences ($< 512$ tokens). The model must recall a fictional location mentioned in a *contextual lead*, then referenced again after several distracting statements. By the time the location needs to be mentioned again after the location prelude, several tokens may have intervened, making it likely that the location tokens may have been evicted. Coarse-grained retrieval schemes risk not finding the entire location as it may be split across pages. This setup mimics conversation-like scenarios. It is especially challenging for token sparsity methods, since prematurely discarding or overlooking the location tokens can irreversibly break the final reference, leading to incorrect or incomplete retrieval of the location name.

We first use GPT-4o-mini to generate 100 fictional location names. We then generate 100 short contextual leads plus matching preludes; then we generate 100 random math, culinary, and philosophical statements. During evaluation, we form 100 sequences adhering to the template. Note that every contextual lead is paired with a matching location prelude. Then, each test sequence is generated as a random sample as follows:

---

**Synthetic Benchmark Template and Sample**

```
<contextual lead>  <location> <philosophical statement> <culinary statement> <math
problem>  <location prelude>  <location>
```
---
```
Shrouded in luminescent fog, ...  color.  The place is:    wraithspire  In the spirit of ...
wisdom waiting to sprout.  Savor the delicate ...  home-cooked love.  If we compute 18 ...
7 gives us 16.    Which location is bathed ...  lights up the shore?    wraithspire
```

---

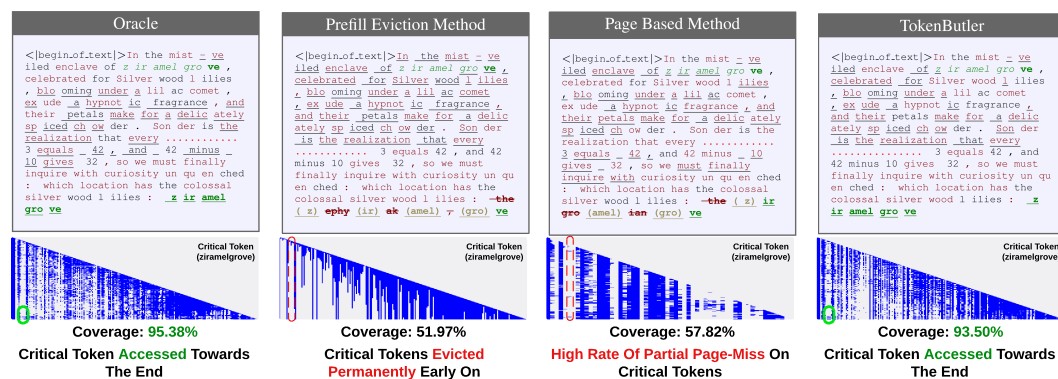

Figure 5: Sample behavior of different KV-Sparsity methods on our synthetic co-reference resolution task. TokenButler outperforms prefill eviction and page-based methods that have clear failure modes due to permanently dropping tokens, or fetching tokens with page-size granularity respectively.

| Model | Oracle | | Token Eviction | | Page-Based | | TokenButler | |
|---|---|---|---|---|---|---|---|---|
| | Acc. | Cov. | Acc. | Cov. | Acc. | Cov. | Acc. | Cov. |
| Llama-3.2-1B | 49.00 | 84.32 | 1.00 | 32.50 | 0 | 19.78 | **49.00** | **82.70** |
| Llama-3.2-3B | 81.00 | 95.38 | 10.00 | 51.97 | 6.00 | 57.82 | **78.00** | **93.50** |
| Llama-2-7b-hf | 77.00 | 93.32 | 18.00 | 57.93 | 1.00 | 34.35 | **78.00** | **94.00** |
| Llama-3.1-8B | 77.00 | 93.47 | 3.00 | 37.50 | 0 | 46.98 | **73.00** | **91.90** |

Table 2: Accuracy and coverage (%) of different KV-sparsity methods on our synthetic dataset. TokenButler outperforms eviction and page-based methods, and approaches Oracle performance.

Since every head may evict tokens based on their importance, we present the attention map for the first head of the 3rd layer (a random choice) in Figure 5. We observe there as well as in Table 2 that (i) prefill eviction methods, e.g. $H_2O$, have low accuracy because they permanently evict older tokens (the location name) once new context is being decoded. (ii) page-based methods, e.g. Quest, **very often** lose part of the location name if it straddles a page boundary in this context-dense example. *Coverage* gives a more detailed view on accuracy. Accuracy is binary, and locations are multiple tokens long, therefore, coverage counts the number of correctly-predicted tokens. For example, if the provided location is 4 tokens long, and a method gets 3 of those tokens correct, it is scored 0.75 in coverage and 0 in accuracy. We see that token eviction and page-based methods are still able to correctly predict around $30 - 50\%$ of the tokens, but not all of them, leading to low accuracy. Table 2 summarizes the results on our synthetic benchmark set on different Llama models.

## 4.3 ACCURACY EVALUATION ON STANDARD BENCHMARKS

We compare with several key works, such as H2O, SnapKV, StreamingLLM and Quest under a uniform token sparsity setup (applied to all layers except the first). We impose a token budget proportional to the input length (e.g. 50% sparsity retains half the tokens). In real-world generative use-cases, new tokens stream in while older tokens remain potentially important, whereas token-eviction based methods like H2O and SnapKV must decide at each step which tokens to discard. Meanwhile, TokenButler and Quest estimate token importance inexpensively for the full input without needing eviction, so they stay efficient even when preserving all tokens.

To compare these approaches fairly, we simulate *token-by-token* decoding on the entire input to simulate generative tasks for standard benchmarks. This implies not having a prefill phase, and requiring H2O and SnapKV to apply their token eviction method at each step, rather than having access to the entire prefill attention map before generating a few tokens for the answer. This provides a more difficult task for token sparsity (for all methods equally) and more closely matches generative use-cases. It also tests whether TokenButler and Quest truly identify *and retain* the right tokens over the full sequence. Furthermore, we evaluate on perplexity and downstream tasks, revealing how token eviction can drop crucial context if done prematurely, and evaluating learned token importance metric in TokenButler.

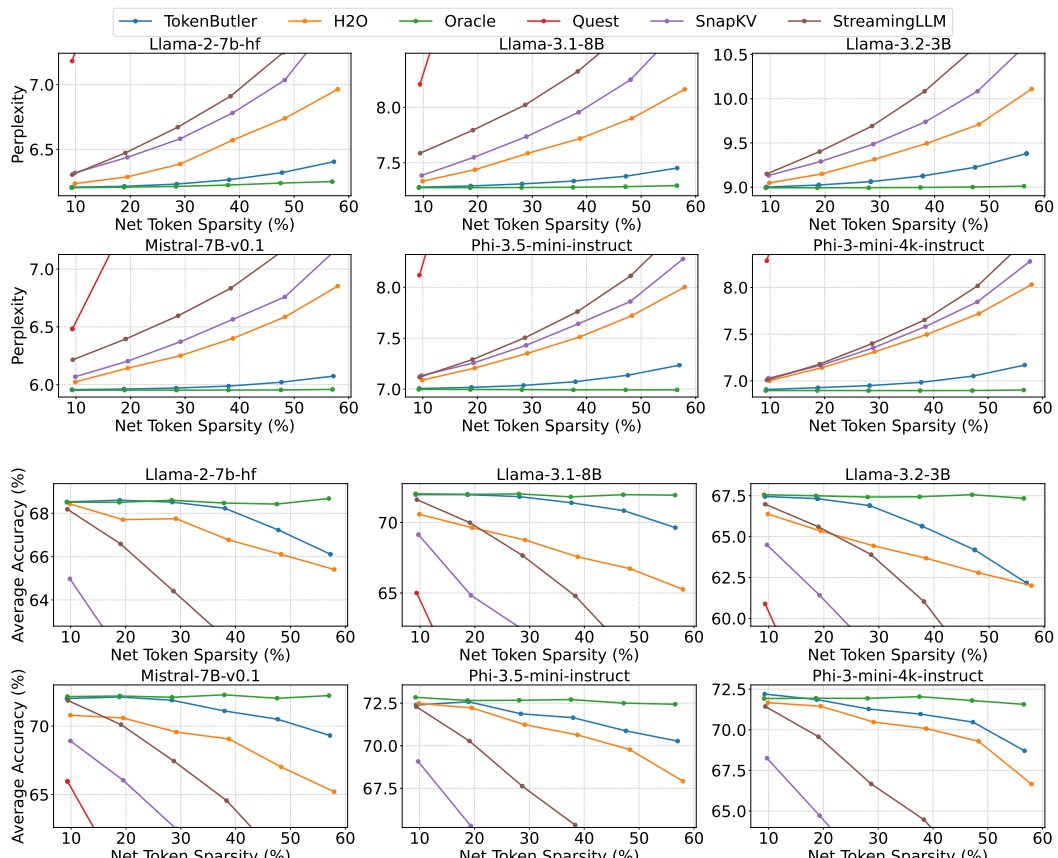

Figure 6: We evaluate TokenButler and other baselines in a uniform token pruning setting. We treat the entire input sequence as a *decode* task, and fix KV-Cache budget as a percentage of the *decoded sequence length*. Net Token Sparsity indicates the average observed token sparsity across all heads, with 4 anchor tokens and no sliding window attention. TokenButler outperforms other metrics at identifying critical tokens. $H_2O$ and SnapKV evict crucial tokens during the decode simulation, and Quest incurs a high rate of page-misses.

| Method (Token Budget) | Qasper | GovReport | QMSum | MultiNews | TREC |
|---|---|---|---|---|---|
| Dense (4096) | 40.23 | 33.09 | 24.3 | 25.21 | 72.5 |
| Oracle (1923) | 36.5 | 31.0 | 23.5 | 25.5 | 72.5 |
| TokenButler (1852) | **32.5** | **30.0** | **23.0** | 24.5 | **72.5** |
| $H_2O$ (2048) | 19.96 | 0.78 | 1.55 | 15.97 | 41.0 |
| TOVA (2048) | 30.14 | 26.15 | 19.7 | **25.04** | 56.5 |
| SnapKV (2048) | 31.37 | 27.03 | 19.93 | 24.97 | 59.0 |

Table 3: Long-context evaluation on Llama-3.2-3B-Instruct with *calibrated sparsity* (Section 4.5). Qasper/TREC: Acc. (%); GovReport/QMSum/MultiNews: ROUGE-L.

StreamingLLM relies on *recency* as a guiding metric, maintaining attention only on the last $W$ tokens within a fixed sliding window while discarding all older tokens, SnapKV on the other hand, determines token importance based on a rolling attention score magnitude, calculated over a small observation window of size 16/32 tokens. In our setup, this observation window is used solely for computing token importance but is not actively maintained unless the tokens it contains are deemed important by the metric itself. Similarly, H2O also employs QK-based importance but operates under a different mechanism. Whenever a newly decoded token exhibits high attention magnitude, it evicts the least important token from its cache, provided the cache is full. Lastly, we include an Oracle baseline, which represents the best possible token sparsity achievable given full access to the LLM's

attention logits. While this provides an upper bound on accuracy performance, it does not reduce computational costs, as it requires a full attention pass to measure token importance before discarding unimportant tokens.

Our evaluation is done on perplexity and average of four downstream tasks (HellaSwag, ARC-Easy, PIQA and WinoGrande) in zero-shot settings. Although these tasks are relatively simple, their critical tokens are often scattered across the entire context. Figure 6 shows the results. The *Oracle* baseline discards tokens *after* calculating their importance, and is thus nearly lossless even at 60% sparsity, revealing substantial redundancy. H2O also achieves decent results, but permanently discards tokens deemed unimportant early on, restricting later access when those tokens become relevant. Meanwhile, Quest's page-level metric underperforms on input lengths up to 1024, because a page size of 16 cannot flexibly capture tokens spread throughout the sequence on context dense tasks. By contrast, TokenButler accurately identifies important tokens in a fine-grained, query-dependent manner, consistently outperforming both eviction and page-based baselines in perplexity and downstream accuracy. From Figure 6, we can see that TokenButler in a fine-grained token access setting without prefill token eviction can offer up-to an 8% improvement in perplexity and downstream accuracy. On long-context tasks (Qasper, GovReport, QMSum, MultiNews, TREC), TokenButler exceeds TOVA Oren et al. (2024) by +4.0 points on average while using fewer tokens. Details are summarized in Table 3.

## 4.4 TOKENBUTLER ON REASONING MODELS

Reasoning models have been shown to have extremely long chain-of-thoughts. The generated CoT can significantly slow down decode, as well as cause significant increase in the KV-Cache size, stressing the decode-time memory bandwidth. To reduce the memory-bandwidth overhead of excessive token-loading, we train TokenButler on the `DeepSeek-R1-Distill-Llama-8B` (DeepSeek-AI et al., 2025) model at 1% of the original model size, for 77M tokens using C4-*realnewslike*. We then evaluate TokenButler's perplexity, as well as two tasks from the OpenLLM Leaderboard (Fourrier et al., 2024) (BBH Causal Judgement (Kazemi et al., 2025) and MMLU Pro (Wang et al., 2024)) where the base reasoning model (`DeepSeek-R1-Distill-Llama-8B`) exhibits good performance. From Figure 7, we can see that even at a very aggressive sparsity of 70%, TokenButler is able to preserve accuracy within 1%, and with a 2% increase in perplexity at 50% sparsity, indicating that TokenButler can be used to reduce the memory and compute overhead of per-token decode on reasoning models well.

## 4.5 LEVERAGING TOP-K RECALL

To assess the impact of token-sparsity in a fair setting, we use global *naive* uniform pruning for evaluation. However, in Table 3 and Figure 8, we utilize a lightweight calibration step that redistributes the token budget across heads using per-head, per-layer Top-K recall (referred to as *calibrated sparsity*). We rank (head, layer) pairs by recall on a small calibration-set of data, and map this ordering to per-head keep ratios (low-recall pairs get lower sparsity, high-recall pairs are sparsified more aggressively), clamp to $[\text{keep}_{\min}, 1]$, and renormalize so the average sparsity matches the target. At inference, the predictor still produces per-token scores; we convert these to masks using the calibrated per-head budget rather than a single uniform threshold. This simple procedure aligns

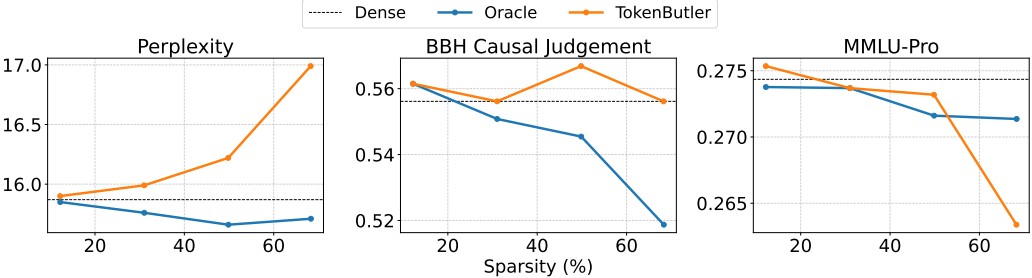

Figure 7: We train TokenButler on the `deepseek-ai/DeepSeek-R1-Distill-Llama-8B` model and evaluate its performance, comparing with a dense baseline and Oracle token pruning.

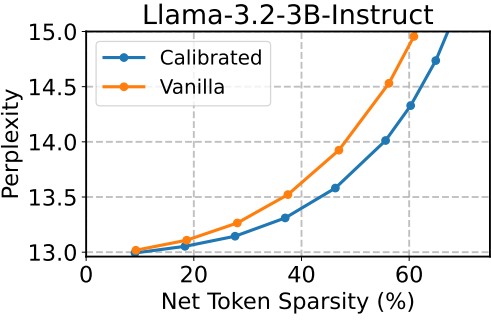

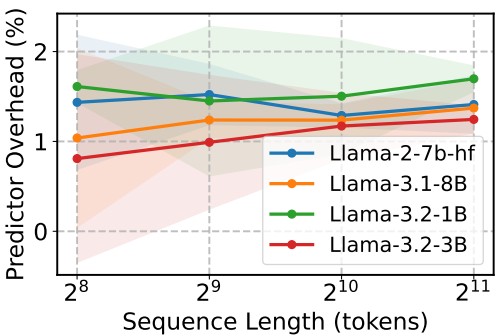

Figure 8: Benefit of re-distribution token budget across heads based on TokenButler's Top-K Recall.

Figure 9: Predictor overhead as a fraction of Llama models on an Nvidia A6000 GPU.

sparsity with the predictor's own error, and improves model quality as shown in Figure 8. Across four Llama variants, the predictor achieves **Recall@1% ≈ 51%**, rising smoothly with K (Appendix §E).

## 5 PERFORMANCE EVALUATION

Despite running alongside the LLM at each decoding step, TokenButler imposes minimal runtime overhead. Figure 9 shows that TokenButler adds roughly 1-2% additional latency. However, this result only quantifies the predictor's own running time in isolation. To evaluate end-to-end performance we integrate Token-Butler with a Llama-3.2 1B model and measure the end-to-end decode throughput under different context lengths in Figure 10. The evaluation utilizes TokenSelect (Wu et al., 2024) code base where we replace their method by a version of TokenButler that predicts the importance per token per layer removing the head dimension from the predictions to match the token retrieval method of the system. Full attention throughput drops as the context length increases, eventually giving an error. Token sparsity methods are needed to counter that. TokenButler throughput is close to the oracle performance and TokenButler is more efficient than TokenSelect as our predictor is very lightweight and does not need to do the dot product with the full original embedding dimension $E$ between Q and K as explained in Section 3.1.

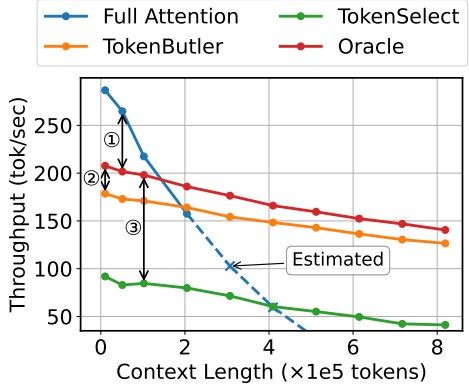

Figure 10: Performance of TokenButler vs. Dense Attention and TokenSelect at 1024 token budget on an H100 GPU. ①: Sparse Attention Overhead. ②: TokenButler Overhead. ③: TokenSelect Overhead.

## 6 CONCLUSION

We present a light-weight predictor that accurately estimates token importance at fine granularity, enabling better token preservation than prior approaches. Our findings suggest that, to handle truly conversational or multi-round tasks, where new text keeps arriving and old tokens can become relevant again, LLMs benefit greatly from *retaining* rather than discarding. When memory limits necessitate a form of compression, it is important to do so in a query-aware, fine-grained manner. Our co-reference experiments show that all-or-nothing eviction or large-page retrieval strategies risk losing important information. TokenButler introduces a light predictor that tracks each head's token preference, preserving the tokens that *actually* matter. This results in up-to 8% gains in perplexity and downstream accuracy. In terms of throughput. We show that TokenButler is very close to the oracle baseline with 10% throughput gap at large context length while outperforming recent methods by up to 3×. Overall, TokenButler paves the way for more precise token management techniques for large language models with minimal performance overhead.

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

## A  APPENDIX

## B  LIMITATIONS

Although TokenButler closes much of the oracle-sparsity accuracy / perplexity gap, our investigation is focused on smaller context-lengths, where we stress-test synthetic natural language multi-round co-referential conversations of $\leq 1K$ tokens. Our primary focus is on demonstrating simple cases where token-eviction methods and page-based token-selection methods can fail. Furthermore, the predictor introduces a modest latency penalty, but at pre-fill still needs to materialize the full L×L attention matrix, as well as retain the complete KV-Cache – costs that can dominate at large token lengths, where token-eviction methods may be useful. Finally, since our focus is on pin-pointing failure modes of low-granularity page-based and eviction-based sparsity methods, our evaluation is limited to downstream evaluation on perplexity and four mid-length benchmarks (PIQA, Winogrande, HellaSwag, ARC-Easy) and synthetic tasks where existing token-sparsity methods already fail.

## C  DISCUSSION

Our experiments demonstrate that *fine-grained, per-head* token importance estimation can improve LLM performance on tasks that require retrieving previously referenced information. A key highlight is the stark difference between TokenButler's high-granularity, query-aware approach and existing token-eviction or page-level compression strategies. Methods like H2O and SnapKV tend to discard tokens prematurely under a size budget, limiting retrieval of critical context later. Page-based approaches (e.g. Quest) are better at retaining old tokens but cannot easily single out individually important tokens, particularly when references straddle page boundaries. Our synthetic co-reference

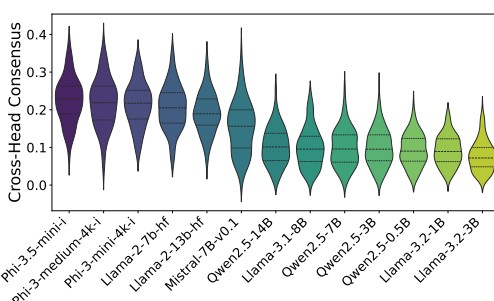
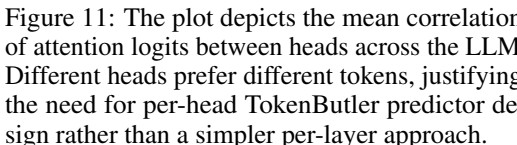

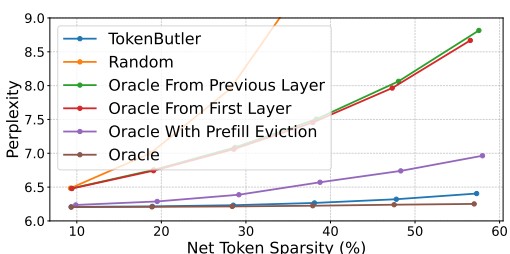

Figure 12: An ablation study on using Oracle token importance from the first layer vs. lookahead (previous layer) for the next layer. Both variants perform worse than Oracle with Prefill Eviction; however, TokenButler and Oracle are still significantly better on `Llama-2-7b-hf`.

Figure 11: The plot depicts the mean correlation of attention logits between heads across the LLM. Different heads prefer different tokens, justifying the need for per-head TokenButler predictor design rather than a simpler per-layer approach.

benchmark highlights these issues: a single location name might be re-invoked well after it appears, yet it can get evicted or split across pages in favor of model performance.

An important observation from Figure 11 is that different heads can have drastically different token preferences. We test cross-head consensus, which is calculated by taking the attention logits from the *last* next-word prediction problem per sequence. We compute the correlation between attention logits across all heads in the LLMs. This gives us a $[NH, NH]$ correlation matrix, and we take the mean of the upper triangular matrix, giving us *mean cross-head* agreement (Cross-Head Consensus) in token preferences. The low correlation observed implies that preserving only a shared subset of tokens selected at the layer level (or from other heuristics) will lead to omission of tokens needed by other heads. TokenButler fixes this by dedicating a Q-K neural network to emulate all heads, ensuring that the tokens each head relies on for context remain accessible. While this slightly increases parameter count (by around 1%), we see a major improvement in perplexity and downstream performance at across token-sparsity levels.

In Figure 12, we first compare Oracle and *Oracle With Prefill Eviction*, which permanently evicts "unimportant" tokens after each next-word prediction. As previously seen, this degrades perplexity, but we also examine whether simpler signals, like reusing attention scores from the *first layer* or the *previous layer*, can guide subsequent layers' token choices without sacrificing tokens. Although such methods do beat a purely random token-dropping baseline, they still do not perform as well as even token eviction strategies. This is because of high cross-head disagreement, which means critical token choices vary widely. Further motivating our design of a decode-focused, fine-grained, per-head token importance prediction system.

## D    EFFECT OF PREDICTOR ATTACHMENT DEPTH

We ablate the layer at which TOKENBUTLER consumes hidden states by training five predictors (each $\approx$54.6M parameters) on `Llama-3.2-3B`, attached at layers $\{0, 4, 8, 16, 24\}$. For target layers 25–27, we evaluate recall across a budget sweep (Recall@$k$%); the resulting curves are shown in Fig. 13. Plotted markers correspond to the measured Recall@$k$% values (e.g., 10/30/50), and lines provide a simple linear interpolation. We find that la ter attachment increases recall across budgets (predictor @24 is best), but layers $< k$ must then become dense, reducing the achievable sparsity budget. In practice, there is a tradeoff such that we (i) attach at a moderate depth to balance recall and sparsity, or (ii) when memory allows, use multiple lightweight predictors (e.g., every 4 layers) to approach the accuracy of attaching at later layers, to retain more sparsity.

## E    PREDICTOR SCALING STUDY

We study how TokenButler's parameter count affects token-importance recovery. All predictors are trained with the same protocol on `Llama-3.2-3B` and evaluated on WikiText2. Table 4 reports Recall@50%, i.e., the fraction of ground-truth high-importance tokens recovered when keeping the predictor's top-50% predictions (averaged over heads and sparse layers).

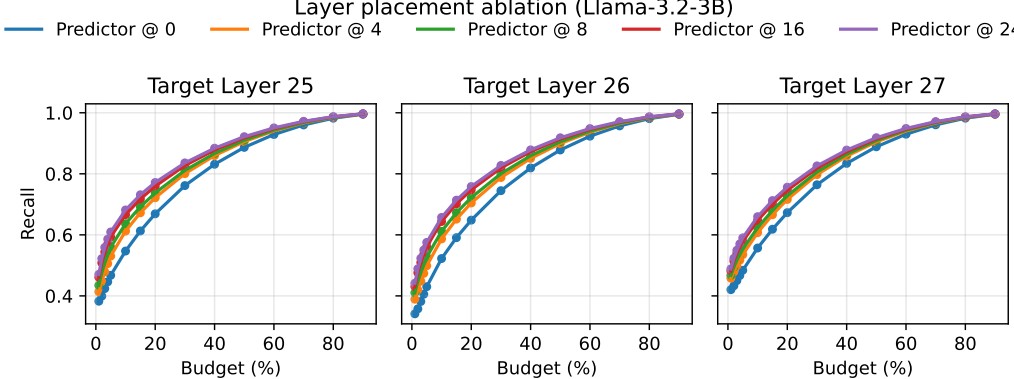

Figure 13: **Layer placement ablation (Llama-3.2-3B).** Recall vs. budget for target layers 25–27. Each curve corresponds to a predictor attached at layers {0, 4, 8, 16, 24}. Markers denote the measured Recall@$k$% points (e.g., 10/30/50). Later attachment (e.g., predictor @24) consistently yields higher recall across budgets, but leaves fewer layers for sparse execution.

| **Predictor size (M params)** | 3.48 | 5.06 | 12.40 | 39.66 | 144.52 | 287.00 |
|---|---|---|---|---|---|---|
| **Recall@50% (%)** | 67.38 | 70.18 | 71.90 | 73.90 | 79.70 | 81.02 |

Table 4: **Predictor size scaling (Llama-3.2-3B).** Larger predictors yield higher Recall@50%.

We observe a smooth scaling trend: increasing the predictor size from 3.48M to 287M improves Recall@50% by +13.6 points (67.38% → 81.02%), providing a convenient accuracy/overhead trade-off for different deployment budgets.

### E.1 TRAINING AND INFERENCE COST.

Training TokenButler does *not* fine-tune the base LLM. On a single A6000 GPU for Llama-3.2-3B, the end-to-end predictor training time scales lightly with predictor size: At inference, the predictor

| Predictor params (M) | 12.4 | 39.7 | 144.5 | 287.1 |
|---|---|---|---|---|
| Time (hh:mm) | 07:17 | 08:01 | 08:32 | 08:42 |

Table 5: Predictor training time on a single A6000 GPU (Llama-3.2-3B).

call adds ∼1–2% wall-clock time in isolation, and end-to-end overhead is ∼12–14% at short contexts, decreasing as context length grows (Table 5).

### E.2 SYNTHETIC CO-REFERENCE BENCHMARK

To rigorously evaluate token sparsity methods under retrieval-intensive scenarios, we developed a synthetic co-reference benchmark utilizing OpenAI's `gpt-4o-mini` model. The benchmark consists of 100 unique fictional location names, 100 paired location introductions and tieback questions, 100 philosophical reflections, 100 culinary descriptions, and 100 short math problems. Each data sample is constructed by randomly selecting one location introduction along with its corresponding tieback question, one location name, one philosophical statement, one culinary description, and one math problem. The resulting sequence is structured such that the location is introduced early in the context, followed by distractor content, and concludes with a prelude statement that prompts the retrieval of the original location name.

This modular generation approach allows for the creation of up to $100^4 = 10^8$ unique sequences by combining different components, ensuring extensive diversity. When a specific number of samples are requested, they are dynamically generated by randomly drawing from the respective pools of location introductions, location names, philosophical statements, culinary descriptions, and math

problems. This on-the-fly sampling methodology ensures that each test instance presents a distinct retrieval challenge, effectively simulating real-world conversational dynamics where important tokens may reappear unpredictably after various interleaved topics. By designing the benchmark in this manner, we specifically target the capability of token sparsity methods to accurately retain and retrieve critical tokens between substantial contextual noise, thereby providing a robust assessment of their effectiveness in maintaining model performance on co-referential tasks.

We provide some *example* texts, and how they perform for different methods (Oracle, Prefill Eviction, Page Based and TokenButler) in the next few pages of the appendix. Each token is separated by a space for clarity. The tokens in *red* are the incorrectly predicted tokens at the last decode stage. The *underlined* tokens are the ones that the token-sparsity method decided to *preserve* when trying to decode the first token of the location word. The first occurence of the location tokens is in green, and their final appearance is bolded in green. Where the method failed to predict the tokens, their 'predicted' token is provided with a strike-through in red, and the *real* correct token is underlined in yellow in brackets.

### Prefill Eviction Method

<|begin_of_text|>L oom ing above a shimmer ing sea of mist , a labyrinth of bi olum ines cent cavern s spark les like a hidden treasure chest beneath the surface . With each step , the walls pulse with vibrant colors , illumin ating the enchant ing mur als crafted by an unknown artisan from ages past . The place is : myst haven . In the manner of S ø ren K ier ke gaard , one might ponder that existence is not a mere state of being , but a relentless pursuit of authenticity amidst the anxiety of choice -- each decision carving out the essence of the self , like a sculpt or unveiling a figure from un yield ing marble . Thus , we are et ern ally engaged in a dialogue with our own potential , whisper ing truth into the depths of uncertainty .. Pl ump , sa ff ron –inf used Spanish pa ella gre ets the senses with its golden hue , as vibrant bell peppers , sweet peas , and tender m ors els of chor izo inter m ingle among a melody of br iny seafood -- sh rimp , m ussels , and cock les gracefully nestled into a bed of al d ente rice . With each bite , the complex layers of smoked pap rika and lemon offer a taste of sun –d renched coast lines , echoing the spirited gatherings of festive Val enc ian fe asts , where laughter dances through the air like the enticing aroma rising from the pan .. If we calculate  26 + 15 – 9 , is the result 32 ? Indeed , it is  32 because 26 plus _ 15 equals 41 , and subtract ing 9 from 41 gives us 32 .. What mysterious location features a labyrinth of bi olum ines cent cavern s that pulse with vibrant colors and bear enchant ing mur als from an unknown artisan ?:  ~~Myst~~ ( myst) ~~ic~~ (haven)

### Page Based Method

<|begin_of_text|>L oom ing above a shimmer ing sea of mist , a labyrinth of bi olum ines cent cavern s spark les like a hidden treasure chest beneath the surface . With each step , the walls pulse with vibrant colors , illumin ating the enchant ing mur als crafted by an unknown artisan from ages past . The place is : myst haven . In the manner of S ø ren K ier ke gaard , one might ponder that existence is not a mere state of being , but a relentless pursuit of authenticity amidst the anxiety of choice -- each decision carving out the essence of the self , like a sculpt or unveiling a figure from un yield ing marble . Thus , we are et ern ally engaged in a dialogue with our own potential , whisper ing truth into the depths of uncertainty .. Pl ump , sa ff ron –inf used Spanish pa ella gre ets the senses with its golden hue , as vibrant bell peppers , sweet peas , and tender m ors els of chor izo inter m ingle among a melody of br iny seafood -- sh rimp , m ussels , and cock les gracefully nestled into a bed of al d ente rice . With each bite , the complex layers of smoked pap rika and lemon offer a taste of sun –d renched coast lines , echoing the spirited gatherings of festive Val enc ian fe asts , where laughter dances through the air like the enticing aroma rising from the pan .. If we calculate 26 + _ 15 – _ 9 , is the result  32 ? Indeed , it is  32 because _ 26 plus _ 15 equals _ 41 , and subtract ing _ 9 from 41 gives us 32 .. What mysterious location features a labyrinth of bi olum ines cent cavern s that pulse with vibrant colors and bear enchant ing mur als from an unknown artisan ?:  ~~The~~ ( myst) haven

### Oracle

<|begin_of_text|>L oom ing above a shimmer ing sea of mist , a labyrinth of bi olum ines cent cavern s spark les like a hidden treasure chest beneath the surface . With each step , the walls pulse with vibrant colors , illumin ating the enchant ing mur als crafted by an unknown artisan from ages past . The place is : myst haven . In the manner of S ø ren K ier ke gaard , one might ponder that existence is not a mere state of being , but a relentless pursuit of authenticity amidst the anxiety of choice -- each decision carving out the essence of the self , like a sculpt or unveiling a figure from un yield ing marble . Thus , we are et ern ally engaged in a dialogue with our own potential , whisper ing truth into the depths of uncertainty .. Pl ump , sa ff ron –inf used Spanish pa ella gre ets the senses with its golden hue , as vibrant bell peppers , sweet peas , and tender m ors els of chor izo inter m ingle among a melody of br iny seafood -- sh rimp , m ussels , and cock les gracefully nestled into a bed of al d ente rice . With each bite , the complex layers of smoked pap rika and lemon offer a taste of sun –d renched coast lines , echoing the spirited gatherings of festive Val enc ian fe asts , where laughter dances through the air like the enticing aroma rising from the pan .. If we calculate  26 + 15 – 9 , is the result 32 ? Indeed , it is  32 because 26 plus _ 15 equals _ 41 , and subtract ing _ 9 from _ 41 gives us _ 32 .. What mysterious location features a labyrinth of bi olum ines cent cavern s that pulse with vibrant colors and bear enchant ing mur als from an unknown artisan ?:  **myst haven**

### TokenButler

<|begin_of_text|>L oom ing above a shimmer ing sea of mist , a labyrinth of bi olum ines cent cavern s spark les like a hidden treasure chest beneath the surface . With each step , the walls pulse with vibrant colors , illumin ating the enchant ing mur als crafted by an unknown artisan from ages past . The place is : myst haven . In the manner of S ø ren K ier ke gaard , one might ponder that existence is not a mere state of being , but a relentless pursuit of authenticity amidst the anxiety of choice -- each decision carving out the essence of the self , like a sculpt or unveiling a figure from un yield ing marble . Thus , we are et ern ally engaged in a dialogue with our own potential , whisper ing truth into the depths of uncertainty .. Pl ump , sa ff ron –inf used Spanish pa ella gre ets the senses with its golden hue , as vibrant bell peppers , sweet peas , and tender m ors els of chor izo inter m ingle among a melody of br iny seafood -- sh rimp , m ussels , and cock les gracefully nestled into a bed of al d ente rice . With each bite , the complex layers of smoked pap rika and lemon offer a taste of sun –d renched coast lines , echoing the spirited gatherings of festive Val enc ian fe asts , where laughter dances through the air like the enticing aroma rising from the pan .. If we calculate  26 + 15 – 9 , is the result 32 ? Indeed , it is  32 because 26 plus 15 equals 41 , and subtract ing 9 from 41 gives us 32 .. What mysterious location features a labyrinth of bi olum ines cent cavern s that pulse with vibrant colors and bear enchant ing mur als from an unknown artisan ?:  **myst haven**

## Prefill Eviction Method

<|begin_of_text|>B ene ath  the  lumin ous  glow  _of _a _thousand _glitter ing  _stars _, _a _crystall ine _lake  _sh immers _with _an _eth ereal _light _that _dances _like _fire _flies _on _its  _surface . _Sur rounded  _by _mountains _carved _from _colorful _gem _stones _, _whispers  _of _long _-lo st _legends  _echo  _through _the _air _, _ming ling  _with  _the _sweet _fragrance  _of _blo oming _night _flowers . _The _place  _is :  _frost _g _len .  _In _the _manner  _of _K _ier ke _gaard , _one  _might _reflect _: _" _True _existence _begins _not _in  _the _relentless  _pursuit  _of _outward _cert ainties _but  _in _the _brave _plunge  _into  _the _depth  _of _one _'s  _own _despair _-- _where _faith _conce _ives _its _true _st _essence _.". _Pl ump , _sa ff ron _-inf used Spanish pa ella gre ets the senses with its golden hue , _as _vibrant _bell peppers , _sweet  _peas , _and _tender m ors els _of _chor izo _inter m ingle _among _a _melody  _of _br iny seafood -- _sh rimp _, _m ussels _, _and _cock les gracefully nestled _into a _bed _of _al d ente rice .  With each bite , the complex layers _of _smoked _pap rika and lemon offer a taste of sun -d renched coast lines , echoing the spirited gatherings of festive Val enc ian _fe asts , where laughter dances through the air like the enticing aroma rising from the pan ..  If we compute  8 + 15 - 5 , is the result 18 ?  Indeed , it is  18 because 8 plus _ 15 equals 23 , and subtract ing 5 gives us 18 ..  Which location boasts a crystall ine lake whose surface spark les like fire flies beneath a canopy of stars ?:  ~~The~~ ( frost) **y** (g) **len**

## Page Based Method

<|begin_of_text|>B ene ath  _the  _lumin ous  glow  _of _a _thousand _glitter ing  _stars _, _a _crystall ine _lake  _sh immers _with _an _eth ereal _light _that _dances _like _fire _flies _on _its _surface .  _Sur rounded _by _mountains _carved _from _colorful _gem _stones , _whispers _of _long _-lo st _legends _echo _through _the _air , _ming ling  _with  _the _sweet _fragrance _of _blo oming _night _flowers .  _The _place  _is :  _frost _g _len .  _In  _the _manner  _of _K _ier ke _gaard , _one _might _reflect  :  _" _True _existence _begins _not _in  _the _relentless  _pursuit  _of _outward _cert ainties _but _in _the _brave _plunge _into _the _depth _of _one _'s _own _despair _-- _where _faith _conce _ives  _its _true _st _essence _.".  _Pl ump , _sa ff _ron _-inf used Spanish pa ella gre ets _the senses _with _its _golden  _hue _, _as _vibrant _bell peppers , _sweet _peas , _and _tender m ors els _of _chor izo _inter m ingle _among _a _melody  _of _br iny seafood -- _sh rimp , _m ussels , _and _cock les gracefully nestled _into a bed of _al d ente rice .  _With _each _bite , _the _complex _layers _of _smoked pap rika and lemon offer a taste _of _sun _-d _renched coast lines , echoing _the _spirited _gatherings _of _festive _Val enc ian _fe asts , _where _laughter _dances _through _the _air _like _the _enticing  _aroma _rising  _from  _the _pan ..  If we _compute _8 + _ 15 _ _ 5 , _is _the result 18 ?  _Indeed _, _it  _is _18 _because _8 _plus  _ 15 equals _ 23 , _and _subtract ing _ 5 _gives _us _ 18 ..  _Which location boasts a crystall ine lake whose surface spark _les _like _fire _flies _beneath _a _canopy  _of stars ?:  ~~The~~ ( frost) **g len**

## Oracle

<|begin_of_text|>B ene ath the lumin ous glow of a thousand glitter ing stars , a crystall ine lake sh immers _with _an eth ereal _light _that _dances _like _fire flies _on _its  _surface . _Sur rounded  _by mountains carved _from colorful gem stones , _whispers _of _long _-lo st _legends  _echo _through _the _air _, _ming ling _with _the _sweet fragrance  _of _blo oming night flowers .  The place is :  _frost _g _len .  In the manner of K ier ke gaard , _one  _might reflect _: _" _True existence begins _not _in _the _relentless _pursuit _of _outward _cert ainties _but _in  _the _brave _plunge  _into  _the _depth  _of _one _'s  _own despair _-- _where faith _conce _ives _its _true _st _essence .". _Pl ump , _sa ff ron _-inf used Spanish pa ella gre ets _the _senses _with _its golden hue , _as _vibrant _bell peppers _, _sweet  _peas _, _and _tender m ors els _of _chor izo _inter m ingle among _a melody _of _br iny _seafood -- _sh rimp _, _m ussels _, _and _cock les gracefully _nestled _into _a _bed _of _al d ente _rice .  With _each _bite _, _the _complex _layers _of _smoked _pap rika _and _lemon _offer _a _taste _of _sun _-d _renched coast lines _, _echoing _the _spirited gatherings _of festive Val enc ian _fe asts , _where _laughter _dances _through _the _air _like _the _enticing _aroma _rising _from  _the _pan ..  If we compute  8 + 15 - 5 , is the result 18 ?  Indeed , it is  18 because 8 plus _ 15 equals _ 23 , and subtract ing _ 5 gives us _ 18 ..  Which location boasts a crystall ine lake whose surface spark les like fire flies beneath a canopy of stars ?:  **_frost g len**

## TokenButler

<|begin_of_text|>B ene ath _the  lumin ous glow _of _a _thousand _glitter ing _stars _, _a _crystall ine lake  _sh immers _with _an  _eth ereal _light _that _dances _like _fire flies _on _its  _surface . _Sur _rounded  _by _mountains _carved _from _colorful _gem stones _, _whispers  _of _long _-lo st _legends  _echo  _through _the _air _, _ming ling  _with  _the _sweet fragrance  _of _blo oming _night _flowers .  The _place  _is : _frost g len .  _In _the _manner  _of _K _ier ke gaard _, _one _might _reflect _: _" _True _existence _begins _not _in  _the _relentless  _pursuit  _of _outward _cert ainties _but  _in  _the _brave _plunge  _into  _the _depth  _of _one _'s  _own _despair _-- _where _faith _conce _ives _its _true _st _essence _.". _Pl ump _, _sa ff _ron _-inf _used Spanish pa ella gre ets _the _senses _with _its _golden hue _, _as _vibrant _bell peppers _, _sweet _peas _, _and _tender _m ors els _of _chor izo _inter m ingle _among _a _melody _of _br iny seafood -- _sh rimp _, _m ussels _, _and _cock les gracefully _nestled _into _a _bed _of _al _d ente rice .  _With _each bite _, _the _complex layers _of _smoked pap rika _and _lemon _offer _a _taste _of _sun _-d _renched coast lines , _echoing _the _spirited _gatherings _of _festive _Val enc _ian _fe asts , _where _laughter _dances _through the air like _the _enticing aroma rising from the pan ..  If we compute  8 + 15 - 5 , is the result 18 ?  Indeed , _it _is  18 _because 8 plus 15 equals 23 , and subtract ing 5 gives us 18 ..  Which location boasts a crystall ine lake whose surface spark les like fire flies beneath a canopy of stars ?:  **_frost g len**

## Prefill Eviction Method

```
<|begin_of_text|>B ene ath a sky painted in
eternal twilight , hills of lum ines cent moss
und ulate like waves , whisper ing _secrets
_to _the playful shadows of dancing fire flies
.  Cr yst all ine springs bubble _up _, _their
water shimmer ing _with hues that transcend
_the spectrum _, inviting weary _travelers _to
drink from _their enchanted depths .  The place
_is :  aer anth ium .  In the spirit _of Conf
uc ius _, let _us ponder _: " A _journey towards
truth begins not _with certainty _, _but _with
the humility _to embrace our ignorance ; _for
in every _question _lies _the _seed _of wisdom
waiting _to spr out ." .  Golden sa ff ron −inf
used bir y ani emerges _from _the pot , its frag
rant _steam ming ling _with _the warm spices
_of card am om and cloves , each pl ump bas
mat i grain g listening _with _the essence _of
mar inated chicken and tang y pick led lime ;
a dish deeply rooted in South Asian heritage
, it offers a harmon ious dance of sm oky ,
savory _, and z esty notes that transports din
ers across bustling b aza ars and op ulent _fe
asts of ancient kings ..  If we calculate  8
+ 15 − 6 , is the result 17 ?  Indeed , it is
 17 because when we add 8 and _ 15 , we get
23 , and subtract ing 6 gives us 17 ..  Which
location features hills of lum ines cent moss
that whisper secrets beneath an eternal twilight
sky ?:  ̶T̶h̶e̶ ( aer) i̶a̶ (anth) o̶s̶ (ium)
```

## Page Based Method

```
<|begin_of_text|>B ene ath a sky painted in
eternal twilight , hills of lum ines cent moss
und ulate like waves , whisper ing secrets to
_the playful shadows _of dancing fire flies
.  Cr yst all ine springs bubble up , their
water shimmer ing _with hues that transcend
the spectrum , inviting weary travelers _to
drink from their enchanted depths .  The place
is :  aer anth ium .  In the spirit of Conf
uc ius , let _us ponder _: " A journey towards
truth begins not _with certainty _, _but _with
the humility to embrace our ignorance ; for
in every question _lies _the seed _of wisdom
waiting _to spr out ." .  Golden sa ff ron −inf
used bir y ani emerges from the pot , its frag
rant _steam ming ling _with _the warm spices
_of card am om and cloves , each pl ump bas
mat i grain g listening _with _the essence _of
mar inated chicken and tang y pick led lime ;
a dish deeply rooted in South Asian heritage
, it offers _a harmon ious dance of sm oky ,
savory _, _and z esty notes that transports din
ers across bustling b aza ars and op ulent fe
asts of ancient kings ..  If we calculate  8 +
_ 15 − _ 6 , is the result  17 ?  Indeed _, _it
is _ 17 because when _we add _ 8 _and _ 15 ,
_we _get _ 23 , and subtract ing _ 6 gives _us
_ 17 ..  Which location features hills of lum
ines cent moss that whisper secrets beneath an
eternal twilight sky ?:  ̶T̶h̶e̶ ( aer) anth ium
```

## Oracle

```
<|begin_of_text|>B ene ath a sky painted in
eternal twilight , hills of lum ines cent moss
und ulate like waves , whisper ing secrets
_to _the playful shadows of dancing fire flies
.  Cr yst all ine springs bubble _up _, _their
water shimmer ing _with hues that transcend
_the spectrum , inviting weary travelers _to
drink from _their enchanted depths .  The place
is :  aer anth ium .  In the spirit of Conf
uc ius , let us ponder :  " A journey towards
truth begins not _with certainty , _but _with
the humility _to embrace our ignorance ; _for
in every _question _lies _the _seed _of wisdom
waiting _to spr out ." .  Golden sa ff ron −inf
used bir y ani emerges _from _the pot , _its
frag rant _steam ming ling _with _the warm
spices _of card am om _and cloves , _each pl ump
bas mat i _grain g listening _with _the essence
_of mar inated chicken _and tang y pick led lime
; _a dish deeply rooted in South Asian heritage
, it offers _a harmon ious dance _of sm oky ,
savory _, _and z esty notes that transports din
ers across bustling _b aza ars and op ulent fe
asts of ancient kings ..  If we calculate  8 +
15 − 6 , is the result 17 ?  Indeed , it is  17
because when we add 8 _and _ 15 , _we _get _ 23
_, _and subtract ing _ 6 gives _us _ 17 ..  Which
location features hills of lum ines cent moss
that whisper secrets beneath an eternal twilight
sky ?:  _aer anth ium
```

## TokenButler

```
<|begin_of_text|>B ene ath a sky painted in
eternal twilight , hills of lum ines cent moss
und ulate like waves _, whisper ing secrets
_to _the playful shadows of dancing fire flies
.  Cr yst all ine springs bubble _up _, _their
water shimmer ing _with hues that transcend
_the spectrum _, inviting weary travelers _to
drink from _their enchanted depths .  _The place
_is : aer anth ium .  In the spirit _of Conf
uc ius _, let _us ponder _: " A _journey towards
truth begins not _with certainty _, _but _with
the humility _to embrace our ignorance ; _for
in every question _lies _the seed _of wisdom
waiting _to spr out ." .  Golden sa ff ron −inf
used bir y ani emerges _from _the pot _, _its
frag rant _steam ming ling _with _the warm spices
_of card am om _and cloves _, _each pl ump bas
mat i grain g listening _with _the essence _of
mar inated chicken _and tang y pick led lime ;
_a dish deeply _rooted _in South Asian heritage
, it offers _a harmon ious dance _of sm oky _,
_savory _, _and z esty notes that transports
din ers across bustling b aza ars and op ulent
fe asts of ancient kings ..  If _we calculate
8 + 15 − 6 , is _the result 17 ?  Indeed _, it
_is  17 because when we add 8 and 15 , we get
23 , and subtract ing 6 gives us 17 ..  Which
location features hills of lum ines cent moss
that whisper secrets beneath an eternal twilight
sky ?:  _aer anth ium
```

## Prefill Eviction Method

```
<|begin_of_text|>Ve iled in an everlasting
twilight , this realm _is painted with lum ines
cent flowers that bloom under _the soft _glow
_of _a silver sky . _ Rivers _of shimmer ing st
ard ust me ander _through em erald fields ,
whisper ing _secrets _to those _who dare _to
_listen . _ The place _is : _ frost wyn . _ In
the manner _of S ø ren _K ier ke gaard , _one
might ponder that existence _is not _a mere
_state _of _being , _but _a relentless pursuit
_of authenticity amidst _the anxiety _of _choice
-- each _decision carving out _the essence _of
_the _self , _like _a _ sculpt or unveiling _a
figure _from un yield ing marble . _ Thus , we
are et ern ally engaged _in a dialogue _with
_our _own potential , _whisper ing truth into
_the depths _of uncertainty .. _ Golden sa ff
ron -inf used bir y ani emerges _from the pot
, its frag rant steam ming _with _the _warm
spices of card am om and cloves , each _pl ump
bas mat i grain _g listening _with _the essence
_of mar inated chicken and tang y pick led lime
; a dish deeply rooted in South Asian heritage
, it offers a harmon ious dance of sm oky ,
savory , and z esty notes that transports din
ers across bustling b aza ars and op ulent fe
asts of ancient kings .. If we calculate _27 +
15 – 8 , is the result 34 ? Indeed , it is _34
because 27 plus _ 15 equals 42 , and subtract
ing 8 gives us 34 .. Which realm is adorned
with lum ines cent flowers and rivers of shimmer
ing st ard ust that whisper secrets to those who
listen ?:  Frost ( frost) wyn
```

## Page Based Method

```
<|begin_of_text|>Ve iled in an everlasting
twilight , _this realm is painted with lum ines
cent flowers that _bloom under _the soft glow
_of a silver sky . _ Rivers _of shimmer ing
st ard ust me ander through em erald fields ,
whisper ing secrets to those _who dare to listen
. _ The place is : _ frost wyn . _ In the manner
of S ø ren _K ier ke gaard , one might ponder
that existence _is not _a mere state _of being
, _but _a relentless pursuit of authenticity
amidst the anxiety of choice -- each _decision
carving _out _the essence _of _the _self , _like _a
sculpt or unveiling a figure from un yield ing
marble . _ Thus , we _are et ern ally engaged in a
dialogue _with our own potential , _whisper ing
truth into _the depths of uncertainty .. _ Golden
sa ff ron -inf used bir y ani emerges from the
pot , its frag rant steam ming ling _with _the
_warm spices of card am om and cloves , each
pl ump bas mat i grain g listening _with _the
essence _of mar inated chicken and tang y pick
led lime ; a dish deeply rooted in South Asian
heritage , it offers _a harmon ious dance of sm
oky , savory , _and z esty notes that transports
din ers across bustling b aza ars and op ulent
fe asts _of ancient kings .. If we calculate
27 + _ 15 – _ 8 , is _the result _ 34 ? Indeed
, _it _is _34 because _ 27 plus _ 15 equals 42
, and subtract ing _ 8 gives us 34 .. Which
realm is adorned with lum ines cent _flowers
and rivers _of _shimmer ing st ard ust that
whisper _secrets _to _those _who listen ?:  the
( frost) wyn
```

## Oracle

```
<|begin_of_text|>Ve iled in an everlasting
twilight , this realm is painted _with lum ines
cent flowers that _bloom under _the soft _glow
_of a silver sky . _ Rivers _of shimmer ing st
ard ust me ander _through em erald fields ,
whisper ing _secrets _to those _who dare _to
_listen . _ The place is : _ frost wyn . _ In the
manner of S ø ren K ier ke gaard , one might
ponder that existence _is not _a mere _state
_of _being , _but _a relentless pursuit _of
authenticity amidst _the anxiety _of _choice --
each _decision carving out _the essence _of _the
_self , _like _a _ sculpt or unveiling _a figure
_from un yield ing marble . _ Thus , we _are et
ern ally engaged _in a dialogue _with _our _own
potential , _whisper ing truth into _the depths
_of uncertainty .. _ Golden sa ff ron -inf used
bir y ani emerges _from _the pot , its frag
rant _steam ming ling _with _the _warm spices _of
card am om _and cloves , each _pl ump bas mat i
_grain g listening _with _the essence _of mar
inated chicken _and tang y pick led lime ; _a
dish deeply rooted in South Asian heritage , it
offers _a harmon ious dance _of sm oky , savory
, _and z esty notes that transports din ers
across bustling b aza ars _and op ulent fe asts
of ancient kings .. If we calculate _27 + 15
– _ 8 , is the result 34 ? Indeed , it is _34
because 27 plus _ 15 equals _ 42 , _and subtract
ing _ 8 gives us _ 34 .. Which realm is adorned
with lum ines cent flowers and rivers of shimmer
ing st ard ust that whisper secrets to those who
listen ?:  frost wyn
```

## TokenButler

```
<|begin_of_text|>Ve iled in an everlasting
twilight , _this realm _is painted _with lum ines
cent flowers that bloom under _the soft glow
_of a silver sky . _ Rivers _of shimmer ing st
ard ust me ander _through em erald fields ,
whisper ing secrets to those _who dare _to
_listen . _ The place _is : _ frost wyn . _ In the
manner _of S ø ren _K ier ke gaard , one might
ponder that existence _is not _a mere _state
_of _being , _but _a relentless pursuit _of
authenticity amidst _the anxiety _of _choice --
each _decision carving out _the essence _of _the
_self , _like _a sculpt or unveiling _a figure
_from un yield ing _marble . _ Thus , we _are et
ern ally engaged _in a dialogue _with _our _own
potential , _whisper ing truth into _the depths
_of uncertainty .. _ Golden sa ff ron -inf used
bir y ani emerges _from _the pot , _its frag
rant steam ming ling _with _the _warm spices _of
card am om _and cloves , _each pl ump bas mat i
_grain g listening _with _the essence _of mar
inated chicken _and tang y pick led lime ; _a
dish deeply _rooted _in South Asian heritage
, it offers _a harmon ious dance _of sm oky ,
savory , _and z esty notes that transports din
ers across bustling b aza ars and op ulent fe
asts _of ancient kings .. If we calculate _27 +
15 – 8 , is the result 34 ? Indeed , it _is _34
because 27 plus 15 equals 42 , and subtract ing
8 gives us 34 .. Which realm is adorned with
lum ines cent flowers and rivers of shimmer ing
st ard ust that whisper secrets to those who
listen ?:  frost wyn
```

## Prefill Eviction Method

```
<|begin_of_text|>C rad led between swirling s
apphire t ides , this breathtaking haven sh
immers under a kale idos cope sky where _the
_sun paints _the horizon _in vibrant _hues of
_gold _and rose . Myst ical creatures roam
among phosph ores cent flora that emit _a _soft
_glow at twilight , illumin ating secret paths
leading _to _hidden water falls that whisper
secrets _of the anc ients . The place _is :
myst ar idge . In the manner _of S ø ren _K
ier ke gaard , one might _say : _" To exist
_is _to confront deeply personal truths that
demand decision ; yet , in every choice made , _a
horizon not _yet _explored van ishes , revealing
the paradox that our freedom shapes us _even
_as _we define ourselves through constraint ."
. A ste aming plate of P anna C otta g list
ens with _a jewel -ton ed raspberry cou lis that
casc ades over its vel v ety cream , ev oking
the sun -d renched landscapes of Italy ; each
spoon ful dances _on _the tongue with rich
notes of _vanilla and a whisper _of _citrus ,
a perfect end to a rustic meal , paying homage
to centuries of culinary art istry ..  If we
calculate  18 + 22 - 7 , is the result 33 ?
Indeed , it is  33 because 18 plus 22 equals 40
, and when we subtract 7 from 40 , we get 33 ..
Which breathtaking haven is cr ad led between
swirling s apphire t ides and features secret
paths illuminated by phosph ores cent flora ?:
Myst ( myst) ar - (idge)
```

## Page Based Method

```
<|begin_of_text|>C rad led between swirling s
apphire t ides , this breathtaking haven sh
immers under a kale idos cope sky where _the
sun paints _the horizon _in vibrant _hues _of
gold and rose . Myst ical creatures _roam among
phosph ores cent flora that emit _a soft glow at
twilight , illumin ating secret paths leading
_to hidden water falls that whisper secrets _of
the anc ients . The place is : myst ar idge
. In the manner of S ø ren _K ier ke gaard ,
one might _say : _" To exist is _to confront
deeply personal truths that demand decision
; yet , in every choice made , a horizon not
_yet explored van ishes , revealing the paradox
that our freedom shapes us even _as _we define
ourselves through constraint ." . A ste aming
plate _of P anna C otta g list ens with _a
jewel -ton ed raspberry cou lis that casc ades
over its vel v ety cream , ev oking _the sun
-d renched landscapes of Italy ; each spoon
ful dances _on the tongue with rich notes _of
vanilla and a whisper _of citrus , a perfect end
to a rustic meal , paying homage to centuries
of culinary art istry ..  If we calculate  18 +
_ 22 - _ 7 , is the result 33 ?  Indeed , _it
_is _ 33 because _ 18 plus _ 22 equals 40 , _and
when _we subtract _ 7 from _ 40 , we get 33 ..
Which breathtaking haven is cr ad led _between
swirling s apphire t ides and features secret
paths illuminated by phosph ores cent flora ?:
The ( myst) ar idge
```

## Oracle

```
<|begin_of_text|>C rad led between swirling s
apphire t ides , this breathtaking haven sh
immers under a kale idos cope sky where _the
_sun paints _the horizon _in vibrant _hues of
_gold _and rose . Myst ical creatures roam
among phosph ores cent flora that emit _a _soft
_glow at twilight , illumin ating secret paths
leading _to _hidden water falls that whisper
secrets _of _the anc ients . The place _is
: myst ar idge . In the manner of S ø ren
K ier ke gaard , one might say : " To exist
_is _to confront deeply personal truths that
demand decision ; yet , in every choice made , _a
horizon not _yet _explored van ishes , revealing
the paradox that our freedom shapes us _even _as
_we define ourselves through constraint ." . A
ste aming plate of P anna C otta g list ens with
_a jewel -ton ed raspberry cou lis that casc
ades over its vel v ety cream , ev oking _the
sun -d renched landscapes _of Italy ; each spoon
ful dances _on _the _tongue with rich notes
_of _vanilla and a whisper _of _citrus , _a
perfect end _to _a rustic meal , paying homage
_to centuries _of culinary art istry ..  If we
calculate  18 + 22 - _ 7 , is the result 33 ?
Indeed , it _is  33 because 18 plus _ 22 equals
_ 40 , _and when we _subtract _ 7 _from _ 40 ,
we get _ 33 ..  Which breathtaking haven is cr
ad led between swirling s apphire t ides and
features secret paths illuminated by phosph ores
cent flora ?: _myst ar idge
```

## TokenButler

```
<|begin_of_text|>C rad led between swirling s
apphire t ides , this breathtaking haven sh
immers under a kale idos cope sky where _the
_sun paints _the horizon _in vibrant hues of
gold _and rose . Myst ical creatures roam
among phosph ores cent flora that emit _a _soft
glow at twilight , illumin ating secret paths
leading _to _hidden water falls that whisper
secrets _of _the anc ients . The place _is
: myst ar idge . In the manner _of S ø ren
_K ier ke gaard , one might _say : _" To exist
_is _to confront deeply personal truths that
demand decision ; yet , in every choice made , _a
horizon not _yet _explored van ishes , revealing
the paradox that our freedom shapes us _even
_as _we define ourselves through constraint ."
. A ste aming plate _of P anna C otta g list
ens with _a jewel -ton ed raspberry cou lis that
casc ades over _its vel v ety cream , ev oking
_the sun -d renched landscapes _of Italy ; each
spoon ful dances _on _the tongue with rich notes
_of _vanilla _and _a whisper _of citrus , _a
perfect end _to _a rustic meal , paying homage
_to centuries of culinary art istry ..  If we
calculate  18 + 22 - _ 7 , is the result 33 ?
Indeed , it _is  33 because 18 plus 22 equals 40
, and when we subtract 7 from 40 , we get 33 ..
Which breathtaking haven is cr ad led between
swirling s apphire t ides and features secret
paths illuminated by phosph ores cent flora ?:
_mystar idge
```

### E.3 THROUGHPUT ON OTHER GPUS

In Section 5, throughput numbers are shown for H100 GPU. Figure 14 shows the throughput for the A6000 GPU which is less powerful. While the intersection points between different methods differs from the H100 GPU, the same trends can be observed where TokenButler outperforms TokenSelect and Full Attention. It is worth noting that TokenButler outperforms Full Attention at a shorter context length for the less powerful GPU.

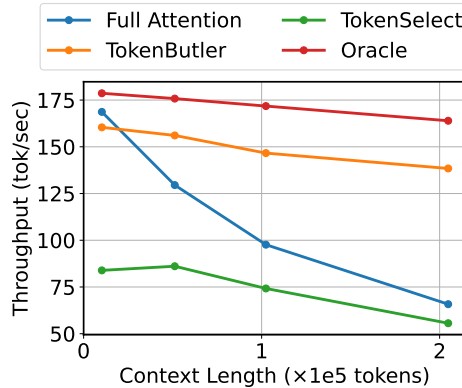

Figure 14: Throughput of TokenButler against full attention and against TokenSelect. The number of tokens selected for sparse attention is 1024 for all. Oracle picks random tokens for performance simulation. Experiment is performed on A6000 GPU.

