# OpenReview forum: "TokenButler: Token Importance is Predictable"
_ICLR.cc/2026/Conference — Submitted to ICLR 2026_

### Official Review · Reviewer_y5HQ · 2025-10-18

**Soundness:** 3
**Presentation:** 3
**Contribution:** 3
**Rating:** 4
**Confidence:** 2

**Summary:**

The paper proposes TokenButler, a learned predictor that make KV caches sparse during decoding. TokenButler estimates the importance of the tokens. The method is plug-and-play at inference, requires modest compute overhead, and targets the decode phase (not prefill). Results show consistent perplexity/accuracy gains.

**Strengths:**

Disclaimer: reviewer is non-expert in the area

1. The paper is well presented.
2. The paper is addressing a real challenging research problem: KV cache cost for long context.
3. The method of the paper is intuitive and novel.
4. The paper has done solid experiments on short to mid context task.

**Weaknesses:**

Disclaimer: reviewer is non-expert in the area

The paper doesn't show experiment results comparing TokenButler with the  token-eviction methods and page-based token-selection methods on long-context cases. In the limitation section, the author explains that this is because his 'primary focus is on demonstrating simple cases where token-eviction methods and page-based token-selection methods can fail.' However, without experiment results for longer context, we simply don't get a complete picture of the actual performance of TokenButler in real case scenarios, especially considering the fact that KV cache cost is the most when the context is long.

**Questions:**

N/A

---

### Official Review · Reviewer_Hj8W · 2025-10-22

**Soundness:** 2
**Presentation:** 1
**Contribution:** 2
**Rating:** 4
**Confidence:** 4

**Summary:**

The paper proposes TokenButler that uses a tiny (<1.2\% extra parameters) predictor to decide which tokens to keep in the KV cache. On a new co-reference benchmark it retains context-critical tokens almost as well as an oracle, letting aggressive sparsity coexist with high accuracy. It lifts WikiText perplexity and downstream task scores by ≥8\% and yields up to 3× higher throughput than prior methods.

**Strengths:**

1. This paper aims to address a timely and important problem
2. Some of the evaluation results are great and insightful
3. The paper analyzes the weakness of eviction strategies (StreamingLLM and H2O) and large-page retrieval strategies (Quest) in the context of co-referential conversation.

**Weaknesses:**

1. The paper claims that token importance is predictable, but does not explain why. (See questions)

2. An important prior study is not discussed. IMPRESS [1] is a training-free adaptive dynamic strategy. It neither evicts tokens nor retrieves pages of tokens. IMPRESS efficiently finds dynamic and query-dependent critical tokens.

[1] Weijian Chen, Shuibing He, Haoyang Qu, Ruidong Zhang, Siling Yang, Ping Chen, Yi Zheng, Baoxing Huai, and Gang Chen. 2025. IMPRESS: an importance-informed multi-tier prefix KV storage system for large language model inference. In Proceedings of the 23rd USENIX Conference on File and Storage Technologies (FAST '25). USENIX Association, USA, Article 12, 187–201.

3. The experimental setting is unclear. For example, the metric "coverage" in Table 2 is defined properly, but the "coverage" in Figure 5 seems to have a different meaning. Moreover, the four tasks (HellaSwag, ARC-Easy, PIQA and WinoGrande) and their accuracy are not explained.

4. The synthetic task is similar to the Needle-in-a-Haystack test, which should be used to evaluate TokenButler. SnapKV performs excellently in this task as shown in the prior paper.

**Questions:**

1. LLMs consist of multiple transformer layers, because different layers capture relations between tokens from different perspectives. How can the single-layer TokenButler predict the importance for all layers, especially when its hidden dimension is reduced to d? As the number of LLM parameters increases (e.g., 13B or 30B), TokenButler may fail to make precise predictions.

2. Figures 3 and 4 show that TokenButler's accuracy is between 70-75\%, which is not high enough. What is the accuracy of prior works?

Minor comments:
- Line 483, In terms of throughput. We -> In terms of throughput, we

---

### Official Review · Reviewer_DXgj · 2025-10-31

**Soundness:** 3
**Presentation:** 3
**Contribution:** 2
**Rating:** 4
**Confidence:** 3

**Summary:**

This paper introduces TokenButler, a lightweight, query-aware predictor intended to identify "critical" tokens in the Key-Value Cache for efficient decoding in large language models. Unlike prior static, eviction-based, or page-based approaches, TokenButler directly predicts per-token, per-head importance ahead of time, using only the first-layer hidden states. The authors train this predictor to match full attention maps and demonstrate its utility in reducing memory and compute bottlenecks, showing improved accuracy and perplexity on standard and synthetic, co-referential benchmarks compared to recent token sparsity methods.

**Strengths:**

The paper addresses a real and timely bottleneck in LLM inference, particularly as context lengths increase and traditional KV-cache management strategies become increasingly inadequate for context-rich tasks. TokenButler's approach—predicting token importance scores for fine-grained, per-head selection using first-layer representations—is clearly motivated and provides a practical balance between computational efficiency and contextual relevance. The technical design is explained with a reasonable level of transparency and thoughtful ablations.

**Weaknesses:**

1.  While the per-token, per-head predictor design is well-motivated, the idea of learning to predict token importance for sparsity and retrieval (from early-layer activations) builds on and is somewhat incremental to recent contextual/pruning/sparsity works, and the overall leap over existing public approaches like TokenSelect, is not as sharply differentiated as the manuscript sometimes suggests. The scope for genuine conceptual innovation is thus moderate.
2.  While Table 3 and Figure 6 include a selection of known sparse attention and token-eviction methods, TokenSelect is not included in the experiments or discussed. As TokenSelect is highly relevant and recent, its absence weakens the claims on comprehensiveness.
3.  A significant weakness is the mismatch between the problem the paper claims to address and the scope of its evaluation.  The introduction explicitly states that LLM contexts have expanded to 128k-1M tokens and that the KV Cache is the primary bottleneck. However, the paper's core accuracy and PPL evaluations Figure 6 are conducted on relatively short context lengths ("up to 1024"). Its novel synthetic task Figure 5 is also limited to "512 tokens".

**Questions:**

1.  Regarding the long-context evaluation, why doesn't the paper compare against other token-level prediction methods, such as TokenSelect, in the main accuracy and perplexity benchmarks? Can the authors also provide results for TokenButler on a standard long-context benchmark like InfiniteBench?

2.  The evaluation would be strengthened by including performance results from applying TokenButler to a wider range of backbone models (beyond those already tested) and evaluating them on other common datasets.

3.  As the results in Figure 5 and 6 is not compelling, could the authors provide more token accuracy results on longer

---

### Official Review · Reviewer_XgYH · 2025-11-01

**Soundness:** 4
**Presentation:** 3
**Contribution:** 4
**Rating:** 6
**Confidence:** 4

**Summary:**

The paper presents TokenButler, a lightweight (~1–1.2% of base model parameters) predictor that estimates token importance per layer and per head using only first-layer hidden states. It predicts low-dimensional query/key pairs whose dot product approximates attention logits, allowing the model to selectively access tokens during decoding without discarding them from the KV cache.

Unlike prior methods that evict tokens (H2O, SnapKV) or rely on page-level retrieval (Quest), TokenButler maintains all tokens in memory but reduces bandwidth by reading only the predicted important ones.

Across several Llama, Mistral, and Phi variants, it achieves 70–75% accuracy for identifying top-50% important tokens and 51% Recall@1% on WikiText2, with minimal overhead. On a synthetic co-reference task, TokenButler approaches oracle performance and outperforms page/eviction baselines. It also improves perplexity and downstream accuracy (e.g., HellaSwag, ARC, PIQA, WinoGrande) by up to 8% at similar sparsity.

Through calibration, TokenButler adapts per-head token budgets, outperforming TOVA, SnapKV, and H2O on long-context tasks such as Qasper and GovReport. The method’s bandwidth savings yield 3× faster decode throughput than TokenSelect. Limitations include full KV retention, dense prefill cost, and small-scale synthetic benchmarks.

**Strengths:**

**1) Clear problem framing and motivation**

The paper clearly explains why query-aware, fine-grained token selection is important and where static or page-level approaches fail. Figures 1 and Table 1 (pp. 2–3) effectively illustrate both the motivation and the limitations of prior methods.

**2) Methodological simplicity and clarity**

Predicting attention for all heads and layers using only first-layer hidden states is a simple and efficient design. This approach aligns with the observed low cross-head agreement (Fig. 11, p. 15), and the architecture in Fig. 2 is easy to implement and conceptually clear.

**3) Broad model coverage.**

Experiments are conducted on multiple model families including Llama-3.2-1B/3B/8B, Llama-2-7B, Mistral-7B, and Phi-3/3.5, as well as a reasoning-oriented model (Sec. 4, pp. 4–5; Fig. 7, p. 8). This variety supports the generality of the findings.

**4) Empirical results that support the main claims.**
Top-K recall and accuracy are strong given the small computational overhead (Figs. 3–4, p. 5).
On the synthetic co-reference benchmark, TokenButler performs close to an oracle and surpasses eviction- and page-based methods (Table 2, p. 6; examples pp. 18–22).
Under uniform-budget simulations, perplexity and downstream accuracy consistently outperform H2O, SnapKV, Quest, and StreamingLLM with up to eight-percent gains (Fig. 6, p. 7; p. 8).
Throughput remains high compared to full attention and TokenSelect (Fig. 10, p. 9).

**5) Comprehensive ablation studies.**

The paper carefully analyzes attachment depth, per-head necessity, predictor size scaling, and per-head budget calibration (Figs. 8, 11, 13; Table 4; pp. 8–9, 15–16).

**6) Transparency about scope and limitations.**

Limitations and trade-offs such as dense prefill cost, full-KV retention, and small latency overhead are clearly discussed (Appendix B, p. 14; Appendix E.1, p. 16), which enhances the paper’s credibility.

**Weaknesses:**

See question

**Questions:**

1) The idea of predictor attachment depth is interesting. Have you explored using multiple predictors attached at different layers simultaneously?

2) In Section 4.5 (Leveraging Top-K Recall), it seems that sparsity is adjusted per head. How does this compare to HEADKV (ICLR 2025) in terms of KV-cache budget management?

3) Is C4-RealNews really the best choice for training data? Could you show at least one more sample or alternative domain?

4) Since the predictor approximates attention, I’m curious how it represents attention sinks. Also, could you elaborate on how this relates to OrthoRank (ICML 2025)?

5) Please provide a clearer analysis of inference benefits, including latency, throughput, and peak memory.

---

### Author Response · Authors · 2025-12-03
**Regarding Long Context Performance**

We would like to thank the reviewers for their thorough review and feedback. In light of the changes to the ICLR review and rebuttal process, to make it easier for the AC, we provide one combined response that addresses all questions reviewers asked.

# Key Question: Long Context Performance

In this paper, our primary focus was to prove that with current static rule-based token sparsity and eviction methods, LLMs fail to faithfully do long-decodes even on sequence length under 1024 tokens. However, we completely understand the concerns raised by the reviewers on long context evaluation. Thus, we train new TokenButler predictor on the Llama-3.1-8B-Instruct model, with a more diverse mix of data. With this, we are able to demonstrate extremely strong performance on even sequence-lengths of 64K. We evaluate our predictor on the RULER benchmark, in two key settings.

In the tables below, we evaluate the Llama-3.1-8B-Instruct model with TokenButler, at a sequence length of 64K. In Single-Turn settings, we provide the context and question together, this lets the prefill happen for the token-eviction method, which means the ‘selection criteria’ is based on the question asked. This lets token-eviction based methods pick ‘the most relevant’ key-tokens and discard the rest.

## In Single-Turn Settings (Llama-3.1-8B-Instruct)

| Method       | Comp. | N-S1  | N-S2  | N-MK1 | N-MK2 | N-MQ  | N-MV  | QA-1 | QA-2 | VT    | FWE   | Avg.  |
|--------|:-----:|:-----:|:-----:|:-----:|:-----:|:-----:|:-----:|:----:|:----:|:-----:|:-----:|:-----:|
| Full Attn   | 1.00  | 100.00| 100.00| 98.96 | 97.92 | 98.96 | 97.66 | 83.33| 59.38| 97.29 | 85.42 | 91.89 |
| StreamingLLM| 8.00  | 15.63 | 12.50 | 13.54 | 13.54 | 14.58 | 17.97 | 56.25| 25.45| 9.38  | 54.10 | 29.35 |
| KVZip       | 8.00  | 100.00| 68.75 | 41.67 | 70.83 | 49.48 | 45.31 | 80.21| 43.75| 86.25 | 82.64 | 66.89 |
| SnapKV      | 8.00  | 100.00| 100.00| 98.96 | 94.79 |100.00 | 97.66 | 83.33| 58.33| 95.00 | 68.75 | 89.68 |
| Quest       | 8.00  | 90.75 | 90.63 | 96.88 | 87.50 | 94.27 | 85.42 | 83.33| 57.29| 77.71 | 81.94 | 84.57 |
| TokenButler | 7.9   | 100.00| 100.00| 100.00| 96.88 | 98.96 | 96.09 | 83.33| 56.25| 90.63 | 77.08 | **89.92** |

However, to focus more on ‘long decode settings’, we evaluate on RULER by first pre-filling the context, and then providing the question in the second turn. This highlights what we discussed in the paper, that token-eviction strategies end up removing tokens that may become critical later in the decode process.

## In Multi-Turn Settings (Llama-3.1-8B-Instruct)

| Method       | Comp. | N-S1  | N-S2  | N-MK1 | N-MK2 | N-MQ  | N-MV  | QA-1 | QA-2 | VT    | FWE   | Avg.  |
|-----------|:-----:|:-----:|:-----:|:-----:|:-----:|:-----:|:-----:|:----:|:----:|:-----:|:-----:|:-----:|
| KVZip        | 8.00  | 100.00| 100.00|  85.42|  84.38| 93.23 | 88.54 | 61.46|51.04 | 93.75 | 81.94 | 83.98 |
| SnapKV       | 8.00  | 58.33 |  25   | 16.67 |2.1    |  24.74| 27.6  | 22.92| 33.33| 26.67 | 65.97 | 30.33 |
| Quest        | 8.00  | 93.75 | 87.5  | 96.88 | 55.21 | 95.57 | 91.93 | 62.5 | 56.25| 89.38 | 83.68 | 81.26 |
| TokenButler  | 7.9   | 100.00| 100.00| 100.00|95.83  | 98.70 | 96.35 |85.42 |57.29 | 90.63 | 76.39 | **90.06** |

In both cases, TokenButler outperforms all prior methods, because it is a learned token-selection strategy and does not rely on static, fixed rules.

## TokenSelect (Llama-3.1-8B-Instruct)

We have added results on long-context benchmark RULER, to compare with TokenSelect. Unfortunately, Table 12 of TokenSelect (LongBench results) does not tell us what the ‘token budget’ is, and this information is also not available in their public code. We have requested the authors of the paper to get more details and will try to add LongBench results for TokenSelect for camera ready. Furthermore, our ‘Dense model’ result for the Llama-3-8B model (the baseline) is significantly better than the one reported in TokenSelect, likely because their evaluation had an OOM error for sequence lengths above 16K. Below, we can see that TokenButler significantly outperforms TokenSelect, InfLLM and StreamingLLM. We see minimal degradation with respect to our Dense baseline at context lengths upto 32K.

| Methods                 | 4K    | 8K    | 16K   | 32K   | 64K   | Avg.      |
|----|-------|------|-------|-------|-------|-----------|
| Llama-3-8B (Dense) | 93.79 | 90.23 | 0.09  | 0.00  | 0.00  | 36.822    |
| StreamingLLM            | 93.68 | 54.48 | 33.77 | 20.35 | 14.88 | 43.432    |
| InfLLM (2K+512)         | 79.79 | 52.43 | 40.12 | 33.60 | 25.68 | 46.324    |
| InfLLM (4K+4K)          | 93.79 | 86.11 | 64.33 | 45.39 | 33.13 | 64.55     |
| TokenSelect (2K+512)    | 93.73 | 82.92 | 71.92 | 65.38 | 59.35 | **74.66**    |
|-----|-------|-------|-------|-------|-------|-----------|
| Llama-3-8B (Dense, Ours)| 96.00 | 94.19 | 94.33 | 94.27 | 91.44 | 94.05     |
| TokenButler (2K+512)    | 95.91 | 94.21 | 91.13 | 93.12 | 88.87 |  92.648   |

---

> ### Author Response · Authors · 2025-12-03
> **Related works, Qwen Long-Context evaluation**
>
> # Other Questions
>
> Below, we address some of the questions the reviewers had
>
> ## Reviewer **Hj8W**
>
> > The paper claims that token importance is predictable, but does not explain why.
>
> A transformer is trained by gradient descent, it learns a fixed, deterministic mapping from inputs to attention patterns and thus outputs. In principle, any sub-computation of this ‘mapping process’ can be approximated by a smaller model. We empirically show that such an approximation can be learned using the early layer activations with predictions accurate enough to drive KV-Cache sparsity and yield real speedups. To put it plainly, it is obvious that token importance is predictable, our contribution is showing that it is learnable enough to be useful for KV-Sparsity and model efficiency.
>
>
> > An important prior study is not discussed. IMPRESS [1]
>
> Thank you for pointing out IMPRESS. It seems they introduce a training-free, importance aware system for multi-tier prefix KV storage. This optimizes disk/CPU/GPU caching for shared prefixes. We believe that the two approaches are complementary, TokenButler can be used as a learned importance estimator inside an IMPRESS-based storage system, potentially giving further reductions in I/O usage and TTFT. We will definitely add this to our related works discussion!
>
> > the metric "coverage" in Table 2 is defined properly, but the "coverage" in Figure 5 seems to have a different meaning.
>
> Thank you for this note, we will try to make it clearer, however, Table 2 and Figure 5 indeed have the same meaning for coverage. As you can see, the Coverage numbers in Figure 5 match the numbers for Llama-3.2-3B in Table 2.
>
>
> > four tasks (HellaSwag, ARC-Easy, PIQA and WinoGrande) and their accuracy are not explained.
>
> We apologize for this, we have cited the papers, which are very well known downstream accuracy evaluation tasks.
>
> ## Reviewer **DXgj**
>
> > the idea of learning to predict token importance for sparsity and retrieval (from early-layer activations) builds on and is somewhat incremental to recent contextual/pruning/sparsity works, and the overall leap over existing public approaches like TokenSelect, is not as sharply differentiated as the manuscript sometimes suggests.
>
> We respectfully disagree, TokenSelect (and other prior methods) are training-free sparsity methods (QK-dot-products, soft-vote across heads). TokenButler is a purely learned metric that aims to, ideally, faithfully match the true attention pattern. While TokenSelect novelty lies in extremely effective systems and kernel design, our aim is orthogonal, to enable ‘metric-free’ token sparsity, by learning to model the true attention patterns more cheaply. This is why we are able to build on the system TokenSelect introduces, and leverage our predictor for higher throughput.
>
> > The evaluation would be strengthened by including performance results from applying TokenButler to a wider range of backbone models
>
> We already tested on Phi, Mistral, Llama and we hope this shows the ‘generality’ of TokenButler. However, we also added the latest model, Qwen-2.5-7B-Instruct model for some long-context tests, as well as used the Llama-3.1-8B-Instruct model in the comparison to TokenSelect. We hope the addition of Qwen backbone is sufficient to prove that the TokenButler predictor can model a sufficiently convincing range of LLM back-bones, as we have demonstrated effectiveness over 4 distinct backbones.
>
> Single-Turn Evaluation at 16K sequence length (Qwen-2.5-7B-Instruct)
>
> | Method       | Comp. | N-S1  | N-S2  | N-MK1 | N-MK2 | N-MQ  | N-MV  | QA-1 | QA-2 | VT    | FWE   | Avg.  |
> |-------------|:-----:|:-----:|:-----:|:-----:|:-----:|:-----:|:-----:|:----:|:----:|:-----:|:-----:|:-----:|
> | Full Attn | 1   | 100.00 | 100.00 | 100.00 | 100.00 | 100.00 | 96.875 | 80.21 | 60.54 | 99.78 | 94.10 | 93.45 |
> | TokenButler | 7.9   | 100.00| 100.00| 100.00| 100.00 | 100.00 | 96.35 |70.83 | 56.25| 100.00 | 90.27 | 91.37 |

---

> > ### Author Response · Authors · 2025-12-03
> > **Predictor placement ablation, training corpus and related works**
> >
> > ## Reviewer **XgYH**
> >
> >
> > > The idea of predictor attachment depth is interesting. Have you explored using multiple predictors attached at different layers simultaneously?
> >
> > To test this idea, we train the Llama-3.1-8B-Instruct model in two settings. One with a single predictor for all layers, and another with a predictor every 4 layers. We train this model on 37 million tokens, at a sequence length of 1024. We measure the Token Hit-Rate at Top-50%, Top-90% and Top-95% tokens.
> > | Predictor Frequency | Hit-Rate @ 50% | Hit-Rate @ 90% | Hit-Rate @ 95%|
> > |-|-|-|-|
> > | Predictor Every 4 Layers | 74.976 | 56.832 | 50.824 |
> > | One Predictor Only | 70.781 | 43.830 | 38.399 |
> >
> > As expected, more predictors improve the hit-rate, however, have 44M more parameters. We further discuss having a single predictor later in the architecture (layer placement) in Figure 13 in the appendix, which may be of interest.
> >
> > > Is C4-RealNews really the best choice for training data? Could you show at least one more sample or alternative domain?
> >
> > To address this concern, we train the Llama-3.1-8B-Instruct model with a new data-mix, which is vastly more diverse than C4-RealNews. We take 100K samples from C4-RealNewsLike, 100K samples from HuggingFaceFW/fineweb-edu, all samples from codeparrot/codeparrot-clean and for better long-context performance, we use samples from RMT-team/babilong, with context sizes of 2k, 4k, 8k and 16k. With this, we find that the Accuracy (ref. Fig 4) improves from 0.72 to 0.75. This proves that the predictor can perform better when trained on more diverse data, however, the core idea is still functional. Thank you for this observation, we will add this ablation to the appendix!
> >
> > > clearer analysis of inference benefits, including latency, throughput, and peak memory.
> >
> > We ran additional analysis on our TokenSelect predictor version used in Section 6 to get a clearer understanding of latency, throughput and peak memory usage as you suggested. We used Llama-3.2-1B-Instruct on a single A6000 GPU for the following analysis. The results show that the predictor improves over TokenSelect  in both throughput and latency as shown in the table below:
> >
> > | Context Length | TokenSelect Throughput (tok/sec) | Predictor Throughput (tok/sec) | TokenSelect P95 Latency (ms) | Predictor P95 Latency (ms) |
> > |----------------|-----------------------------|----------------------------------|--------------------------|------------------------------|
> > | 64K            | 70.8                      | 141.61                           | 16.59                     | 7.12                        |
> > | 128K           | 64.71                       | 134.05                           | 16.66                    | 7.47                         |
> >
> > For peak memory usage, we analyze 2 models with their predictors runtime usage of memory for a sequence length of 64K and top_k = 1024 and we generate 20 tokens. Overhead of the predictor is shown in the following table:
> >
> > | Model                   | Model only Memory (MB) | Model + Predictor Memory (MB) | Predictor Overhead (%) |
> > |-------------------------|-------------------------|--------------------------------|-------------------------|
> > | Llama-3.2-1B-Instruct   | 20,975.65              | 21,167.65                      | 0.92                   |
> > | Llama-3.2-3B-Instruct   | 30,136.65               | 30,440.66                      | 1.01                   |
> >
> >
> >
> >
> > > Since the predictor approximates attention, I’m curious how it represents attention sinks. Also, discussion on HeadKV, OrthoRank
> >
> > Since the objective we train with is MSE over the attention mask vs predicted attention mask, the attention sinks are actually represented similarly to how the base LLM itself represents the sink. The percentage-mass allocated to the first four tokens across 128 examples, was on average 40.30% for our predictor, whereas it was 50.6% for the true-attention.
> > With TokenButler, we aim to use true attention weights, to learn the underlying behavior of the attention mechanism instead of studying heuristic methods. The discussed sink-token-orthogonality shows promising results on perplexity, but comparing Table 8 of OrthoRank, we find that our Qasper accuracy is 40.23% vs the reported 14.08 (even for the dense model). This may be because the -Instruct variant was not used for LongBench in OrthoRank, making direct comparison a bit difficult given lack of accompanying code. We will however discuss this further in the related work, and try to reproduce OrthoRank for the camera ready version! HeadKV is complementary to our predictor, where instead of adjusting sparsity per head based on Top-K Recall, we can use the head-level allocation strategy introduced in that paper along-side our predictor!

---

### Meta-Review · Area_Chair_FtPe · 2025-12-30

**Summary:**

This paper proposes a lightweight, learned token selection mechanism that addresses KV-cache inefficiency in long-context decoding by predicting which past tokens are relevant at each decoding step. These predictions enable fine-grained, query-aware access to the full KV cache, rather than relying on coarse eviction or paging. By avoiding the common failure modes of these traditional strategies, especially in context-rich, co-referential scenarios, the method achieves improvements in both generation quality and decoding throughput, with minimal overhead.

The paper addresses a timely systems bottleneck in long-context LLM inference by proposing a simple, learned predictor that leverages early-layer representations to estimate token importance for fine-grained selection. It presents several strong and insightful empirical results, including a case study that highlights common failure modes of eviction- and page-based KV strategies in context-rich, co-referential scenarios.

**Reviewer Concerns:**

The primary weakness noted by the reviewers is a mismatch between the paper’s motivation and its evaluation scope. While the paper highlights long-context bottlenecks (128k-1m tokens), the main results focus mostly on shorter contexts, so the evidence for the long-context claims is limited. In addition, reviewers point out that the evaluation does not include enough well-matched comparisons against strong recent baselines, nor standard long-context benchmarks or stress tests. One reviewer (DXgj) also questions whether the method is sufficiently novel relative to closely related work. Finally, reviewers mention that some experimental details are not described clearly (e.g., metric definitions and configuration settings), which makes parts of the results harder to follow and to reproduce.

The rebuttal addresses some of the reviewers’ questions, but several key concerns remain. First, the additional long-context evidence remains limited (e.g., results up to 64K tokens), whereas the paper's framing emphasizes much longer contexts (128k-1m tokens). As a result, the submission offers only partial support for its long-context claims, and much of the main-paper evaluation continues to focus on shorter contexts. Second, the reviewers’ requests for more complete comparisons and standard long-context validation are not fully addressed: the paper lacks well-matched baselines such as OrthoRank and HeadKV, and omits common stress tests like needle-in-a-haystack. Furthermore, some experimental reporting (metrics, settings, and budget/throughput details) would benefit from clearer specification to make the results easier to interpret and verify.

**Reviewer Scores:**

Reviewer XgYH already had a positive score and I don't anticipate them changing their score drastically.

Reviewer DXgj's main concern is about experiments on long context tokens. While the new rebuttal experiments work towards this, they would need to be properly examined and perhaps would require another round of review. So I would anticipate this reviewer's score to stay the same.

Reviewer Hj8W questioned the clarity of the description of the experiments and suggested adding a new particular experiment (needle in a haystack, see above). I did not see this sufficiently addressed in the rebuttal.

Reviewer y5HQ was a low confidence reviewer whose opinion should not be weighted heavily.

---

### Decision · Program_Chairs · 2026-01-26

Reject